# Mechanically induced topological transition of spectrin regulates its distribution in the mammalian cell cortex

Andrea Ghisleni[1], Mayte Bonilla-Quintana [2], Michele Crestani [1,3], Zeno Lavagnino [1], Camilla Galli[1,4], Padmini Rangamani [2] ✉ & Nils C. Gauthier [1] ✉

The cell cortex is a dynamic assembly formed by the plasma membrane and underlying cytoskeleton. As the main determinant of cell shape, the cortex ensures its integrity during passive and active deformations by adapting cytoskeleton topologies through yet poorly understood mechanisms. The spectrin meshwork ensures such adaptation in erythrocytes and neurons by adopting different organizations. Erythrocytes rely on triangular-like lattices of spectrin tetramers, whereas in neurons they are organized in parallel, periodic arrays. Since spectrin is ubiquitously expressed, we exploited Expansion Microscopy to discover that, in fibroblasts, distinct meshwork densities co-exist. Through biophysical measurements and computational modeling, we show that the non-polarized spectrin meshwork, with the intervention of actomyosin, can dynamically transition into polarized clusters fenced by actin stress fibers that resemble periodic arrays as found in neurons. Clusters experience lower mechanical stress and turnover, despite displaying an extension close to the tetramer contour length. Our study sheds light on the adaptive properties of spectrin, which participates in the protection of the cell cortex by varying its densities in response to key mechanical features.

The plasma membrane is a responsive composite material where the lipid bilayer and a plethora of attached cortical proteins create a continuously adapting interface with the extracellular environment. Among these attachments, the actin-spectrin meshwork constitutes a ubiquitous scaffold critical in preserving cell shape and integrity, extensively studied in red blood cells (RBCs) and neurons[1,2]. In erythrocytes, the meshwork imposes the peculiar biconcave shape and acts as the unique dissipator of shear forces that RBCs experience in the bloodstream. Pathogenic spectrin variants affect the meshwork integrity and result in hemolytic diseases that underlie the fundamental role in RBCs integrity and protection against mechanical cues[3,4]. In neuronal axons, spectrin acts as a ruler that spatially

organizes critical membrane components such as ion channels and cortical adapters (i.e., ankyrins), while pathological variants are linked to neurodevelopmental and neurodegenerative conditions[5,6]. The physiological importance of spectrin is also supported by murine knockout models presenting embryonic lethality due to neurological and cardiovascular defects[7].

At the structural level, the building block of this meshwork consists of tetrameric head-to-tail $(\alpha\beta)_2$-spectrin dimers[8]; these rod-shape tetramers are cross-linked by short actin filaments (~ 35 nm in length) via a pair of actin-binding domains (ABD) present at each end of the β subunits. Hierarchical meshwork organization is obtained by the binding of multiple spectrin tetramers to a unique short actin filament,

[1]IFOM ETS, The AIRC Institute of Molecular Oncology, Milan, Italy. [2]Department of Mechanical and Aerospace Engineering, University of California San Diego, La Jolla, CA, USA. [3]Present address: Laboratory of Applied Mechanobiology, Department for Health Sciences and Technology, ETH Zürich, Zürich, Switzerland. [4]Present address: Humanitas Cardio Center, IRCCS Humanitas Research Hospital, Rozzano (Milan, Italy. ✉e-mail: prangamani@ucsd.edu; nils.gauthier@ifom.eu

which functions as a connecting node[9]. Of the different paralogue genes expressed in humans (2 α and 5 β), only the pair αI-βI display the restricted expression pattern to RBCs. Pioneering work by electron microscopy unveiled a triangular-like lattice organization of spectrin in erythrocytes[10], where 5−7 spectrin tetramers are connected via a single actin node. Although discrepancies in lattice dimensions were reported depending on the sample preparation method, this conformational arrangement long stood as the unique spectrin organization described, until the advent of optical super-resolution microscopy and the identification of a periodic actin-spectrin array with regular spacing (~180−190 nm in length) along axons[11]. Alternative ultrastructural spectrin topological organizations in other cellular lineages have not been described. Outside of the Central Nervous System and RBCs, heterogeneous spectrin distribution in epithelial cells and the accumulation at the intercalated disk in muscle tissue have been observed at optical microscopy resolution[12,13]. We recently reported similar observations in many other cell types[14]. Recent work in erythrocytes using super-resolution approaches also suggests a more heterogeneous distribution with a discrete organization of nanoscale clusters[15]. Those observations support the possibility that spectrin might create discrete mesoscale cortical domains of different densities (in the order of 1−10 μm² scale) with specific and specialized topological organizations and functions that remain to be elucidated.

Dynamically, key molecular features have been extrapolated from in vitro protein mechanics[16,17]. The unfolding/refolding capacity of spectrin repeats, triple-helix domains arranged in series that constitute the central rod of the protein, have led to the conclusion that spectrin and related proteins (i.e., actinin, utrophin, nesprin, etc.) might possess stretch/relaxation capabilities upon mechanical cues also in vivo. However, a direct translation of those mechanical capabilities in a cellular context remains elusive, and attempts to elucidate these mechanisms have been limited again to the neuronal and erythroid backgrounds. Indeed, a role for spectrin as a shock-absorber in neurons has been described[18], following evidence in nematodes where spectrin stabilizes sensory neurons against mechanical deformation[19]. In erythrocytes, the actin-spectrin meshwork is the only structural cytoskeleton[4]; therefore, this cell system represents the most simplistic model to study the meshwork mechanical properties and its interplay with the plasma membrane. As for the topological organizations, the mechanical context of the meshwork is unexplored in other cellular systems that largely express spectrin. To measure forces sustained by cytoskeletal components and to map the mechanical stress landscape at the subcellular level, different genetically encoded FRET-based sensors have been designed. Krieg et al. hypothesized a constitutive mechanical tension in the spectrin meshwork in peripheral axons of nematodes, being able to measure a local decrease in stress upon laser axotomy procedure[20]. Orientation-based FRET sensor inserted in αII-spectrin has been developed as a proxy for estimating cortical forces in eukaryotic cells[21,22]. Given the hypothesized role as a load-carrying spring or tension absorber, it is rational to ask whether these heterogeneous distributions and mechanical states of spectrin may act in concert with the complex hierarchical and dynamic organization of the cell cortex.

Here, we identified a critical role for spectrin in the regulation of cell area and membrane trafficking by adopting different densities. By Expansion Microscopy, we unveiled that mammalian fibroblasts can harbor spectrin clusters configured into parallel arrays with a periodicity of ~180−190 nm, forming a pattern fenced by actin stress fibers where spectrin is at low tension and low turnover. Cluster formation is driven by actomyosin organization and contractility, and challenged during mechanical perturbations. On the contrary, low-density spectrin displays higher tension and turnover, with heterogeneous but diffused organization. Using dynamic computational simulations that integrate topology, force changes, and actomyosin, we examined under which conditions spectrin undergoes topological and mechanical transitions to regulate its distribution under the plasma membrane.

## Results

### Spectrin forms high-density clusters between actin stress fibers

We reported that spectrin is dynamically organized in mesoscale cortical domains (1−10 μm² in size) complementary to the actin cytoskeleton during cell spreading, and excludes endocytic pits by a local shutter mechanism[14]. Here, we sought to determine spectrin ultrastructural organization in fully adherent mouse embryonic fibroblasts (MEFs) that exhibit a mature cytoskeleton. Diffraction-limited TIRF microscopy (TIRFM) investigations highlighted discrete zonal compartmentation of endogenous βII-spectrin, which formed high-intensity clusters without colocalization with F-actin (Fig. 1A). These clusters were preferentially found between ventral stress fibers and never observed in cell protrusions. Analyzing the distribution of fluorescence intensity on a μm² scale revealed a log-normal distribution of βII-spectrin, not observed for F-actin (Fig. 1B, Supplementary Fig. 1A and Supplementary Table 1). We concluded that different and discrete zonal organizations by heterogeneous spectrin densities might coexist in the cortex of fibroblasts.

Spectrin clusters (defined as $P_{0.95}$ of the intensity distribution, Supplementary Fig. 1B) were segmented for both βII-spectrin and αII-spectrin (2.08 ± 34 and 2.18 ± 3 μm² area respectively), while clusters of bigger size were identified for F-actin (6 ± 73 μm², Fig. 1C). Moreover, the aspect ratios (AR) for both spectrins were reminiscent of an elliptic shape (AR: 1.6 ± 0.6 and 1.7 ± 0.7 for βII-spectrin and αII-spectrin), compared to the AR of F-actin which resulted more elongated as expected for long fibers (AR: 2.3 ± 2, Fig. 1D). A matching alignment between F-actin dominant direction with both βII-spectrin and αII-spectrin emerged, but with lower coherency for spectrins (Fig. 1E, F, see "Methods" for orientation analysis). These results indicated that spectrin clusters have the tendency to align along the direction dictated by F-actin, the cytoskeletal structure with higher coherency.

### Spectrin participates in membrane homeostasis in fibroblasts

We observed the mutual exclusion between spectrin, actin, and clathrin pits at the plasma membrane. Notably, we observed the exclusion of endocytic pits at high-density spectrin zones, supporting the role of spectrin as a dynamic membrane organizer[14]. To address the role of spectrin in membrane homeostasis, genome-edited *sptbn1* KO MEFs were generated (Supplementary Fig. 2A). Spectrin-depleted clones exhibited an overall reduction in the projected cell area compared to control cells, accompanied by an increase in clathrin-mediated endocytosis monitored by transferrin uptake (Fig. 1G, H and Supplementary Fig. 2B−C). The specific removal of high-density clusters is challenging, particularly by genetic approaches (i.e., genome-editing, RNA interference, etc.). However, our previous and present results, in combination with recent pre-print literature[23], support the global membrane homeostatic role of spectrin in regulating both cell shape (area) and membrane trafficking (endocytosis) also in fibroblasts. Moreover, spectrin has been shown to embrace a mechanoprotective role in red blood cells and neurons[20,24]. To address whether spectrin offers similar contributions in fibroblasts, membrane fragility upon osmotic stress was measured (by CellTox™ Green intensity), and *sptbn1* KO MEFs displayed an increase in fragility when exposed to hypotonic but not hypertonic conditions (Supplementary Fig. 2D and Supplementary Table 2).

### Periodic topology is found in high-density spectrin clusters

Since spectrin mesoscale density at the plasma membrane fluctuates and affects membrane homeostasis, we aimed to disentangle the topological arrangements of spectrin, how clusters form under the membrane, and how they can potentially exclude other membrane structures such as endocytic pits[14]. By STED microscopy, we resolved

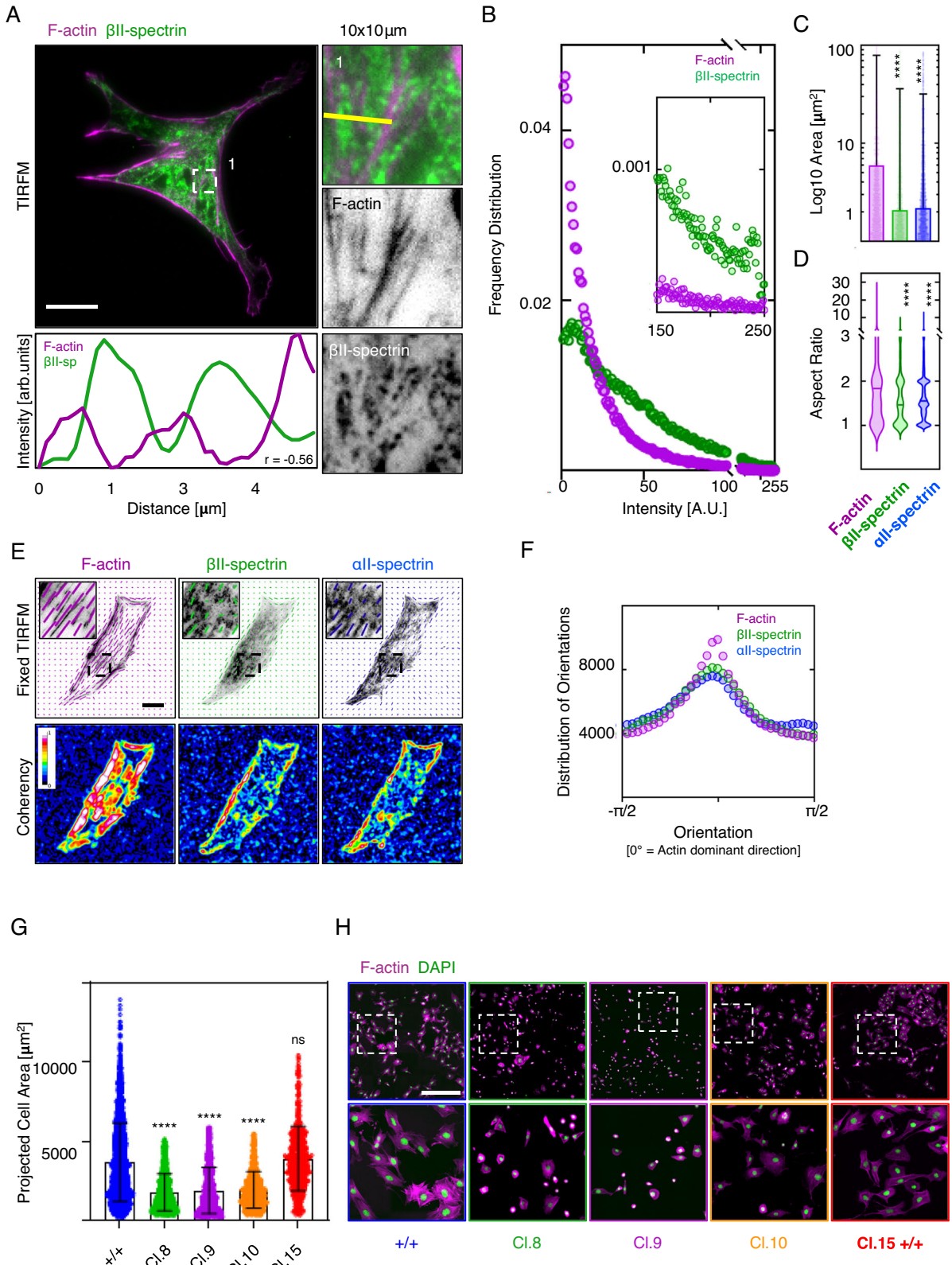

spectrin organization with the expected periodicity in axons (Supplementary Fig. 3A). As shown in HeLa[25], the same approach failed to show sufficient improvements in resolution on the diffuse spectrin immunostaining displayed by MEFs (Supplementary Fig. 3B, E). To bypass this limitation, we implemented Expansion Microscopy (ExM)[26]. This approach ensured a consistent ~4x expansion factor compared to the original size of the biological specimen (Fig. 2A, B and Supplementary

Fig. 4A) and resolved erythroid spectrin lattice organization with results comparable to the ones reported by conventional super-resolution approaches (Supplementary Fig. 4D)[27].

To our surprise, in fibroblasts, βII-spectrin clusters presented a topological organization reminiscent of the periodic pattern identified in axons (Fig. 2C, D and Supplementary Fig. 4E). In line with the TIRFM results (Fig. 1A), clusters were found in the flat ventral zone of the cell

**Fig. 1 | Spectrin clusters align between actin stress fibers. A** MEFs immunolabelled for βII-spectrin (green) and F-actin (phalloidin magenta), imaged by total internal reflection microscopy (TIRFM, scale bar 20 μm). Overlay and single-channel images are shown, as well as zooms related to the square box (1). Images are representative of at least 3 independent experiments. **B** Frequency distribution of signal intensities per μm². Inset highlights the different signal distributions of F-actin (magenta) and βII-spectrin (green) at high-intensity zones (*n* = 40 cells). **C** Area of high-intensity clusters (P$_{0.95}$ of signal intensity distribution) and related aspect ratio (**D**) highlights the smaller and elliptic shape of βII-spectrin and αII-spectrin clusters compared to elongated F-actin (data are presented as mean ± SD, statistical analysis Brown-Forsythe and Welch ANOVA test with multiple comparisons, ****p < 0.0001). **E** Orientation analysis of endogenous βII-spectrin (green), F-actin (phalloidin, magenta), and αII-spectrin (blue). The vectors highlight the orientation of the signal in the local window for the three independent channels. Coherency heatmaps are shown (LUT 16-colors, scale bar 20 μm). **F** Distribution of orientations of βII-spectrin (green), F-actin (magenta), and αII-spectrin (blue), normalized to F-actin dominant direction (*n* = 40 cells). **G** Projected cell area of clonal populations KO for βII-spectrin (*sptbn1*), based on the phalloidin staining presented in **H** (scale bar 500 μm, *n* = 3307(+/+), 1202(Cl.8), 1712(Cl.9), 886(Cl.10) and 527(Cl.15) cells in 3 independent experiments, data are presented as mean ± SD, statistical analysis Brown-Forsythe and Welch ANOVA test with multiple comparisons, ****p-value < 0.0001).

rather than in protrusions, and located in the cortical region constrained by ventral stress fibers (stained by β-actin antibody, Fig. 2C). Epitope inter-distance was calculated without the need of additional image deconvolution approaches and the extrapolated values (180-to-190 nm, Fig. 2D) resembled the periodicity identified in neurons (Supplementary Fig. 4E)[11]. As shown in Fig. 2C–E, our immunofluorescence microscopy investigations also support the parallel orientation of the spectrin periodic array to the stress fiber axes. The ~ 4x resolution gain obtained by ExM also resulted in a significant reduction in labeling density (Supplementary Fig. 4A). As a result, no intermingled actin staining was observed between βII-spectrin puncta, while the conventional phalloidin labeling is known to not resist the gelation and expansion steps of the protocol. We speculated that short actin filaments that connect spectrin tetramers are immunologically secluded in spectrin clusters or do not contain the specific β-actin isoform immunolabelled here. On the other hand, the commercial antibody against αII-spectrin recognizes two specular epitopes on the tetramer that were not resolved by ExM; a clear periodic organization was not obtained for αII-spectrin (Supplementary Fig. 4B), neither its localization with respect to the βII-spectrin one (epitopes mapping is shown in Fig. 2A). However, dense αII-spectrin regions were found in correspondence with dense βII-spectrin counterparts, and spatial cross-correlation analysis confirmed the bona fide of the antibody immunoreactivity (Supplementary Fig. 4C). Our results unveiled a non-random organization of spectrin clusters, as opposed to the less defined organization found in the rest of the fibroblast cortex (zooms 1 and 2, Fig. 2E). When βII-spectrin Nearest Neighbor Distance (NND) measurements were performed in adjacent regions within the same cell, only a mild decrease was observed between the diffused (860 ± 250 nm) and the constrained configurations (704 ± 192 nm). However, NND analysis reports the proximity between local maxima; at our resolution, nearest epitopes in dense spectrin regions should not be assumed as directly connected as well as the possibility of underestimating the number of nearest maxima for a non-single molecule imaging approach needs to be accounted for. Taken together, these results raise the intriguing possibility that the diffused spectrin meshwork might pack up reversibly in the form of periodic clusters by applying different topological and mechanical constraints (Fig. 2G).

## Actomyosin dynamics controls the formation and disassembly of the spectrin clusters

We asked whether these spectrin clusters were dynamic. Following actin and spectrin by TIRFM, we observed the formation and dissipation of GFP-βII-spectrin clusters fenced between stress fibers on the ventral side of fibroblasts (Fig. 3A). As cells changed shape and moved, clusters followed the framework created by the stress fibers (Fig. 3A, zooms 1–6 and Supplementary Movie 1). Those clusters, as well as actin stress fibers, were successfully segmented with the approach previously implemented (Supplementary Fig. 1B), and autocorrelation coefficients between consecutive frames were calculated; we observed a higher tendency for GFP-βII-spectrin to auto-correlate over time

compared to RFP-actin (0.80 ± 0.07 and 0.53 ± 0.12 respectively, Fig. 3B, C). While accurate shape descriptors (i.e., size, AR) for those clusters were difficult to obtain from live TIRFM datasets, it was possible to analyze the distribution of orientations and relative coherency of GFP-βII-spectrin and RFP-Actin signals. As observed for the endogenous staining, GFP-βII-spectrin predominantly aligned with the direction dictated by the more coherent actin cytoskeleton (Fig. 3D, E), and sub-cellular regions defined by high βII-spectrin coherency correlated with high actin coherency (Fig. 3D, asterisk). Altogether, these results confirmed the conclusions drawn from fixed TIRFM and ExM investigations corroborating the more passive nature of spectrin clusters compared to the actin cytoskeleton, which is highly dynamic during cell motility.

To explore how morphological constraints may regulate spectrin cluster formation, we engineered the substrate and seeded fibroblasts on microfabricated fibronectin-coated lines (4 μm in thickness), spaced by 12 μm of non-adhesive surface. On these instructive surfaces, fibroblasts were forced to sprout out thin and elongated processes similar to the proximal portion of axons (Fig. 4A, B). ExM on those specimens highlighted the tendency of spectrin to create elongated clusters parallel to the axis of the adhesive lines (Fig. 4A, B and Supplementary Fig. 5A). However, differently from the axons, βII-spectrin clusters failed to consistently organize into a periodic pattern along the entire length of the cell protrusions, indicating that additional molecular or cell-specific determinants were required to reach the full barrel and periodic organization described in neurons.

Given the technical challenges of studying the functional contribution of spectrin topologies by genetic approaches (see results of *sptbn1* KO MEFs), we aimed to drive spectrin clustering by the administration of different cytoskeletal-affecting compounds. Pharmaceutical stabilization of F-actin has been implemented to obtain enhanced ultrastructural details of the spectrin mesh and actin braids in axons[28]. Among the different compounds screened to qualitatively assess the effect on the spectrin meshwork (Supplementary Fig. 6A, B), we identified the actin turn-over inhibitor Jasplakinolide and the myosin inhibitor Blebbistatin as the most insightful. We treated fibroblasts with 100 nM Jasplakinolide for 3-4 h to stabilize the actin cytoskeleton without blocking actomyosin contractility[29]. As a consequence, we observed a strong concentration of spectrin signal by the formation of large clusters (Fig. 4C and Supplementary Movie 2). When fluorescent signal coherency across the cell was analyzed over time, anticorrelated behaviors between GFP-βII-spectrin and RFP-actin were observed only under Jasplakinolide treatment. Specifically, the decrease in signal coherency observed for actin when stress fibers were disassembled, was accompanied by the formation of large βII-spectrin clusters with increased coherency (Supplementary Fig. 5B). ExM showed that large clusters were formed by multiple smaller patches of periodic patterns resembling the ones previously observed, resulting in increased βII-spectrin signal autocorrelation (Fig. 4D and Supplementary Fig. 5C, D). These results suggest that, under Jasplakinolide perturbation, small clusters initially fenced by stress fibers

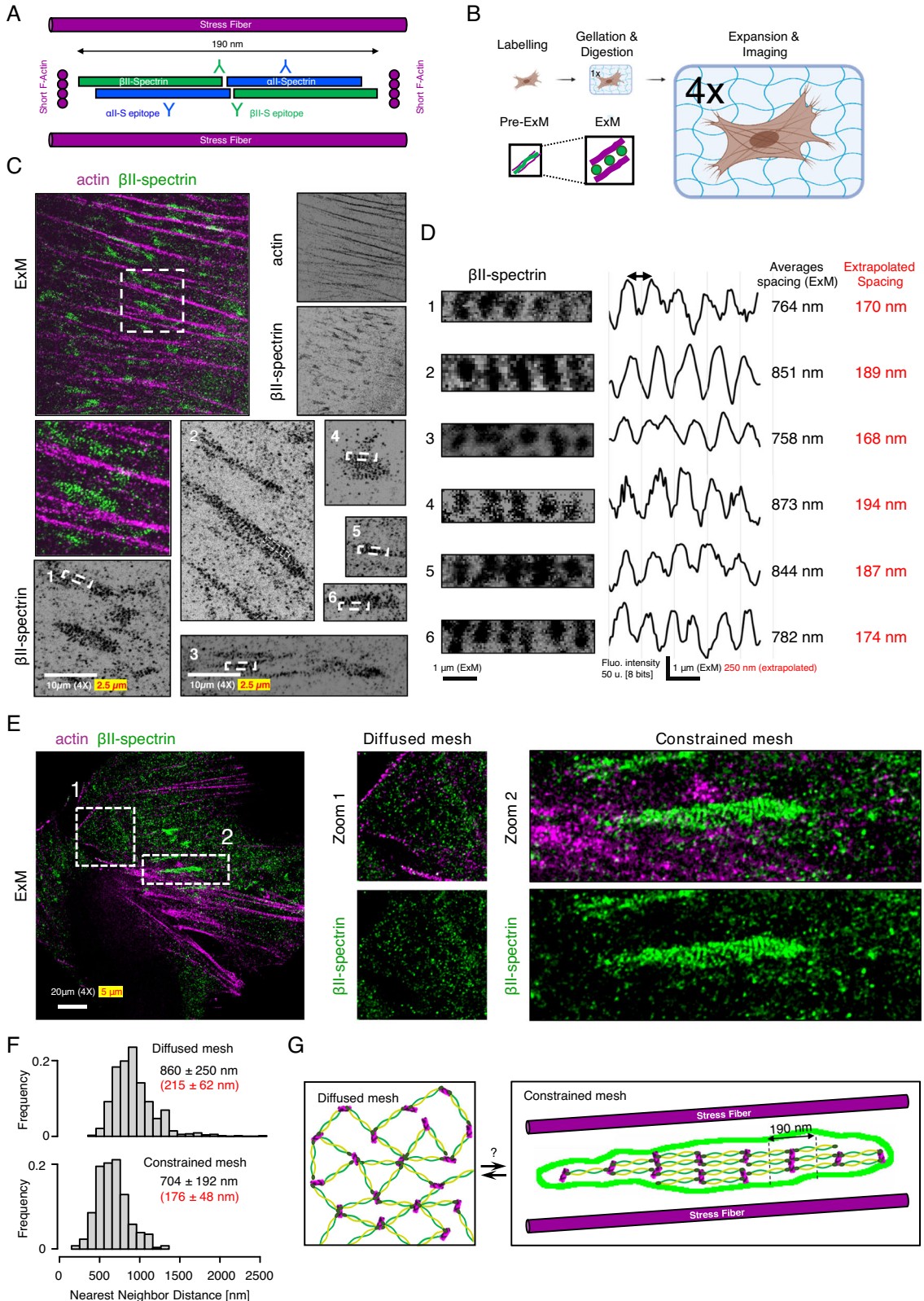

might be dynamically aggregated by myosin contractility. This leads to the formation of large spectrin clusters made of smaller arrays without dominant orientation, potentially due to the lack of the main directional axis imposed by the stress fibers. When this periodicity followed a clear local directionality, line scan analysis confirmed the expected βII-spectrin periodic spacing of ~800 nm, the same we identified in untreated fibroblasts (Supplementary Fig. 5D). These critical

observations support the concept that the spectrin meshwork does not coalesce randomly but instead, clusters in a controlled manner to form an ultimately packed and ordered array of ~180 nm parallel tetramers. This concept of controlled coalescence of spectrin driven by actin organization is also reinforced by our results using Latrunculin A or Cytochalasin B, that, as opposed to Jasplakinolide (and to some extent Cucurbitacin E), are completely affecting actin organization

**Fig. 2 | Expansion Microscopy unveils the periodic organization of spectrin clusters. A** Cartoon representation of actin-spectrin tetramers (estimated length ≈ 180/190 nm) and the relative positions of the immuno-reactive epitopes (Y symbols). **B** The experimental pipeline to enhance the spatial resolution by Expansion Microscopy (ExM). **C** Representative ExM images of MEFs immunolabelled for βII-spectrin (green) and β-actin (magenta) are shown; in white is reported the "real" scale bar of the image (10 μm) and in red the extrapolated scale according to the expansion factor of 4x. The zooms related to βII-spectrin white boxes (1–6) are shown to highlight periodic clusters (**D**), and the corresponding intensity line scans

and peak-to-peak distances are reported (real in white and extrapolated in red). **E** Representative ExM image showing βII-spectrin (green) in the diffused (zoom 1, white dashed box) and constrained mesh configurations between stress fibers are shown (zoom 2). Images are representative of at least 3 independent experiments. **F** Nearest Neighbor Distance distributions in the two configurations are calculated for the βII-spectrin epitopes. Adjacent zones within the same cells can display large discrepancies in terms of spectrin organizations (drawn in **G**). Panel (**B**) was created with BioRender.com and released under a Creative Commons Attribution-NonCommercial-NoDerivs 4.0 International license.

rather than stabilizing it and, as a consequence, showed formation of large clusters without proper alignments (Supplementary Fig. 6).

To gain insights into the molecular mechanisms that can drive the clustering of the spectrin meshwork, we measured the effects of Jasplakinolide on protein dynamics by FRAP and observed a reduction in mobility and increased half-time recovery for GFP-βII-spectrin in treated compared to untreated fibroblasts (Fig. 4E and Supplementary Table 3). The strong negative effect on RFP-actin dynamics was also confirmed by the more than 90% reduction in mobility (Fig. 4F and Supplementary Table 3). Interestingly, the effect induced by Jasplakinolide could be reversed, and the cells could recover their shape and motility behavior after washout of the compound (Supplementary Fig. 7A, B), indicating the reversibility of the process and the cell viability despite the strong morphological alterations. On the other hand, myosin-II inhibition by Blebbistatin had opposite effects and promoted cluster dissipation (Fig. 4C, D and Supplementary Figs. 5E, 7C). As reported by us and others[14,30], Blebbistatin decreases GFP-βII-spectrin dynamics (Fig. 4E). Normalized cluster area was measured during drug washout experiments (Supplementary Fig. 7D–F). Spectrin clusters dissolved after hours from Jasplakinolide washout (Supplementary Fig. 7E), while the reaction to Blebbistatin washout followed a much faster kinetic; cluster area abruptly increased after reawakening of myosin contractility, accompanied by a decrease in cluster size and a return to homeostatic levels during the time course of the experiment (Supplementary Fig. 7F). Altogether these results support the model that envisions myosin contractility as the main homeostatic driver in spectrin dynamics, required for spectrin clustering into the ultimate periodic organization. This might represent an elegant and non-stochastic mechanoadaptive and mechanoprotective mechanism to store spectrin reservoirs.

### Spectrin is in a low stress and low turnover state in the clusters

The spectrin mesh is historically considered as an elastic scaffold that determines the shape of the associated plasma membrane, based on in vitro evidence of stretch-and-recoil capabilities upon mechanical stress possessed by the numerous spectrin repeats that form the central rod of the molecule[16,17]. Considering all the results presented here, these molecular features raise the intriguing possibility that different spectrin organizations might emerge from different tensional landscapes of the cell cortex. To investigate whether this was the case, we envisaged a genetically encoded intramolecular FRET-based βII-spectrin sensor similar to the stress sensor developed for αII-spectrin and nesprin[22,31]. Since only the β subunits of the (αβ)₂ tetramer harbor the actin-binding domain (ABD), inserting the cpst FRET pair immediately after the "neck" that bridges the ABD with the central rod might provide a more direct strategy to investigate spectrin tensional state compared to the original cpst-αII-spectrin. Moreover, this approach offers the possibility to apply the headless construct lacking the ABD as a stress-insensitive reporter (Fig. 5A). We measured FRET sensitized emission to derive the inverted FRET ratio (invFRET) as proposed in the original report: a lower ratio equals to lower stress and vice versa (see "Methods"). Based on the results presented here, we assumed that spectrin clusters characterized by high fluorescence intensities might present a lower tensional state (low invFRET) compared to regions of

the cell characterized by lower fluorescence intensities (measured in the acceptor channel by Venus intensity, Fig. 5B). We therefore transiently expressed in fibroblasts cpst-βII-spectrin full length (FL) or the headless ΔABD, and applied the same drug treatments previously described (Fig. 4C) to promote different spectrin organizations as means to map the correlation between mechanical stress and local spectrin density. At the whole-cell scale, no correlation was identified between the mean invFRET ratio and Acceptor intensity (Supplementary Fig. 8A); overall the resulting invFRET was not biased by the transfection efficiency of the FRET sensor, which can differ between independent cells and experiments. Moreover, the drug treatments applied did not lead to degradation of both the endogenous and the transfected cpst-βII-spectrin FL/ΔABD (Supplementary Fig. 8B, C), excluding a contribution of proteolysis in FRET results. Semi-automated analysis was performed at μm² resolution (Fig. 5C). While all the experimental conditions displayed normal frequency distribution of invFRET values, only in Jasplakinolide treated cells a bimodal distribution and higher coefficient of variation emerged (CV, defined as the ratio of the standard deviation to the mean), with a specific population of values characterized by low invFRET ratio (low tension). The same population did not appear in the stress-insensitive cpst-βII-spectrin-ΔABD and overall, no significant changes were detected between control and Blebbistatin treatments. Interestingly Blebbistatin treated cells displayed the lowest CV value (0.282), indicating the less dispersion of values around the invFRET mean, and thus a more homogenous tension when contractility was inhibited. We concluded that when treated with Jasplakinolide, a portion of the spectrin meshwork shifted to a state characterized by lower mechanical stress. To correlate mechanical stress and meshwork density, scatter plots with color-coded Kernel density distributions were generated to visualize the relationship between these two parameters. The population characterized by low invFRET and high fluorescence intensity only appeared when cells expressing cpst-βII-spectrin-FL were treated with Jasplakinolide, and not in all the other experimental conditions tested (Fig. 5C, red boxes and bars). Given the observations presented by ExM and FRAP under Jasplakinolide treatment (Fig. 4D, E and Supplementary Fig. 5D), we indirectly extrapolated that periodic spectrin clusters corresponded to the regions characterized by lower turnover and lower mechanical stress, and vice versa (see Fig. 5D as a cartoon representation of the differential configurations of the spectrin mesh upon drug perturbations). Osmotic shocks, hypotonic in particular, are known to alter cell mechanics by challenging membrane and cortical components[32,33]. We measured the spectrin cluster area in a protocol of stretch-relaxation by fractional osmolarity cycles (Supplementary Fig. 8D, E, 1x-0.5x-1x, see Supplementary Table 5 for media compositions). Interestingly, the GFP-βII-spectrin cluster area decreased sharply after the infusion of the hypotonic media, reaching an equilibrium state. Actin was also reacting but with a smaller amplitude. These results suggested that spectrin clusters reacted to cortical challenges by dissipating, helping the cell to maintain membrane integrity as we observed in Supplementary Fig. 2D. We also observed the partial reformation of spectrin clusters upon return to iso-osmotic media, indicating the reversibility of the mechano-adaptive response of spectrin to osmotic challenges.

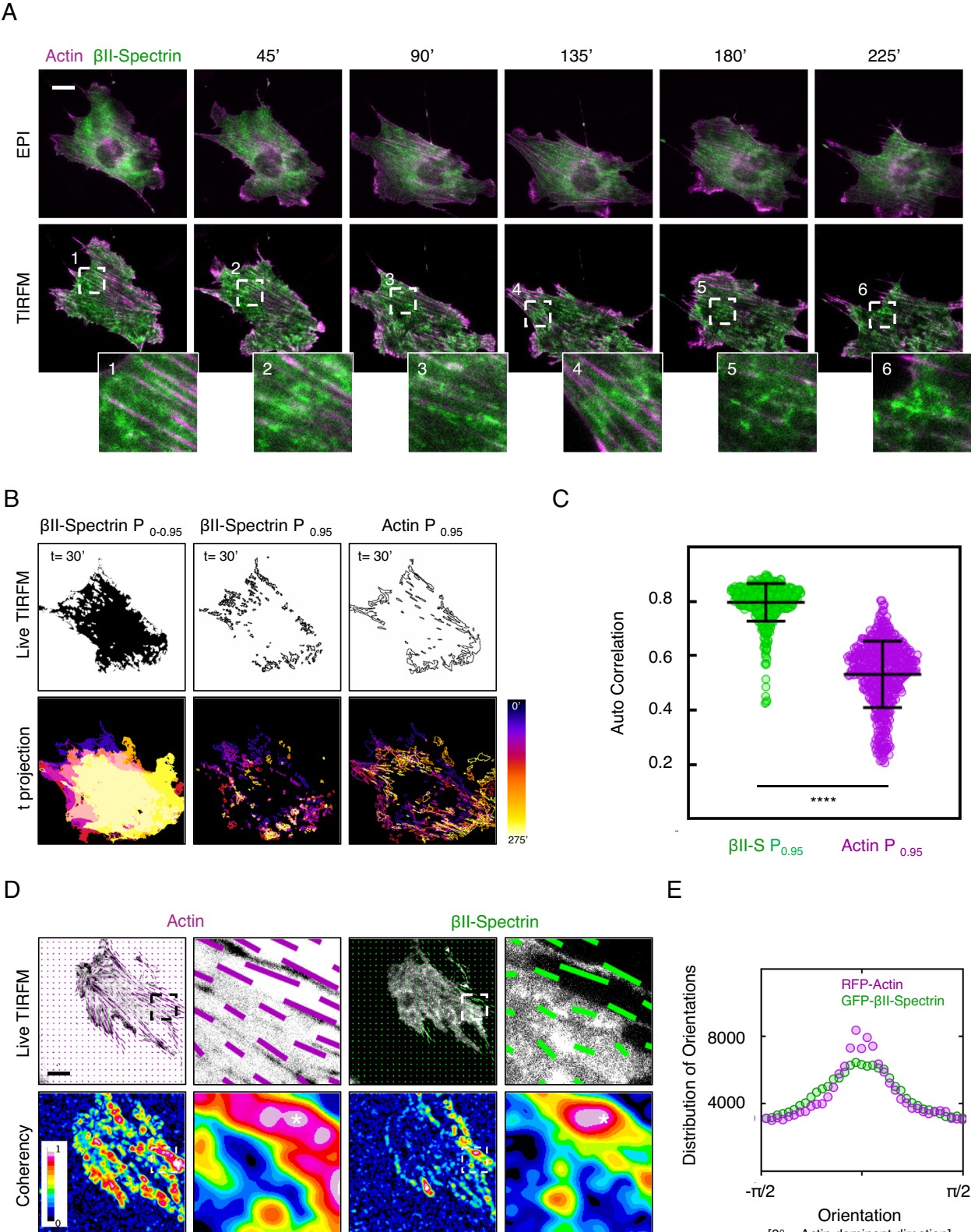

Theoretical model examining conformational changes of spectrin arrays highlight the need of spectrin bundle detachment and active constraint by stress fibers

To investigate how topological constraints can give rise to spectrin clustering and the related mechanical properties, we developed a computational model. In this mechanical model, the different cytoskeletal elements, namely, filament bundles (tetramers) and crosslinkers are represented by edges and nodes, respectively[34–39]. The edges behave like springs generating energy when their length differs from a resting length, or cables increasing their energy with length. This energy generated by the edges is transmitted to the nodes connecting them, thereby changing their position (Fig. 6A, B, see "Methods"). By calculating the position of the nodes due to the force generated by the energy of the edges at each time step, we examined

**Fig. 3 | Spatio-temporal evolution of the spectrin clusters. A** Live imaging in dual mode (EPI and TIRF microscopy) of MEFs transiently transfected with GFP-βII-spectrin (green) and RFP-actin (magenta). Relevant frames are shown (scale bar 20 μm), as well as dynamic zooms of high-intensity βII-spectrin clusters only observed in the TIRF plane (white dashed boxes). Live images are representative of at least 3 independent experiments. **B** Temporal analysis of GFP-βII-spectrin clusters evolution ($P_{0.95}$ of fluorescent signal intensity distribution). The resulting mask is shown with the clusters outlined, while the color-coded temporal projection is shown for the same cell presented in A, to highlight the dynamic nature of these clusters over time. **C** Auto-correlation coefficients between subsequent frames related to GFP-βII-spectrin (green) and RFP-actin (magenta) are shown ($n = 8$ independent cells and 486 frame pairs analyzed, data are presented as mean ± SD, statistical analysis paired t-test, ****$p$-value < 0.0001). **D** Representative frames during time lapse analysis of MEFs transiently transfected with GFP-βII-spectrin (green) and RFP-actin (magenta). The vectors highlight the orientation of the signal in the local window for the two channels. Coherency heatmaps are shown. **E** The graph reports the distribution of orientations of GFP-βII-spectrin (green) and RFP-actin (magenta), normalized to the Actin dominant direction ($n = 8$ cells in independent time lapses).

the transition between different configurations (Supplementary Fig. 8F). First, we considered a network composed of spectrin bundles (edges) arranged in a triangular lattice linked by short actin filaments (nodes). We chose a triangular lattice as initial configuration because it is isotropic and more suitable for tiling a 2D space. Therefore, it avoids clustering of the edges. Moreover, it resembles the known spectrin configuration in red blood cells. Such a network does mechanical work when the length of its edges is smaller or larger than the resting length of 180 nm (Supplementary Fig. 8F, Supplementary Movie 3). This potential energy exchange produces a restorative force that dictates the movement of the short actin filaments until the edges recover their resting length, and thus, their force is minimized (Fig. 6C, Supplementary Fig. 8G and Supplementary Movie 3).

We hypothesized that when external forces are applied to the network, the arrangement of the edges changes to a clustered configuration. To test this hypothesis, we applied an external force to a relaxed spectrin network by connecting it to fixed nodes through cables. These fixed nodes mimic focal adhesions connecting the cell cytoskeleton to the substrate, while cables represent stress fibers (Fig. 6D). The cable element exerts a contracting force. Because we assumed that focal adhesions are anchored to the substrate, the spectrin network stretches in the horizontal axes (Fig. 6E and Supplementary Movie 4). We observed that the resulting network has high mechanical stresses because the diagonal spectrin bundles are stretched and the vertical bundles are compressed. To reduce the mechanical stresses, we introduced spectrin unbinding at a low strain, i.e., removing compressed bundles, as suggested in ref. 40, which results in a less stressed configuration (Fig. 6F and Supplementary Movie 4). In such a configuration, the closer the short actin filaments are to the top and bottom of the network, the fewer spectrin bundles they connect, hinting that in the neuronal-like configuration, fewer bundles are connected by short actin filaments than in the extended configuration (Fig. 6F).

Next, we investigated what happens when the spectrin mesh is constrained by stress fibers. To do so, we included stress fibers with a spring and cable element attached to focal adhesions in their ends (Fig. 6G). We mimicked the forces generated by actin polymerization in the stress fibers by imposing a greater length in the edge connecting the stress fiber to a focal adhesion[41]. Such a network evolves to a lower-stress configuration (Fig. 6H and Supplementary Movie 5), where the spectrin bundle orientation is qualitatively more similar to experimental observations (compare with Fig. 1A, E and Fig. 2C, E). However, stress fibers are active and are known to move toward each other[42]. We mimicked this motion by moving the focal adhesions toward each other at a constant speed for the first half of the simulation and fixing them thereafter. Figure 6I (and Supplementary Movie 5) shows that the resulting network relaxes further and that the movement of stress fibers reduces the number of spectrin bundles connected to a short actin filament.

In conclusion, our model shows that the clustering of the meshwork with lower stress is mediated through the detachment of spectrin bundles with a lower strain, and by active stress fibers. As depicted in Fig. 2G, the reduction of the number of spectrin bundles per short actin filament promotes neuronal-like periodicity in the constrained mesh that otherwise would exhibit a shorter periodicity. However, this

change is not transmitted to the center of the spectrin network. Since our experimental observations pointed to a critical role of myosin contractility in driving spectrin topological transition, we sought to investigate and derive key positional and dynamic parameters from cellular observations to implement myosin workload into this theoretical model.

## Local pulsatile myosin dynamics remodel spectrin-rich cortical domains

The contribution of myosin-II-dependent contractility in regulating spectrin dynamics is instrumental for the shape regulation of erythrocytes[30,43]. We have reported a reduction in spectrin dynamics also in fibroblasts treated with Blebbistatin[14]. Here we observed that myosin leads to the formation of large clusters of periodic topology when actin dynamics are impaired (Jasplakinolide treatment). By contrast, blocking myosin contractility while keeping actin dynamics (Blebbistatin treatment) leads to homogenous spectrin density distribution without clusters. In addition, washout of those drugs leads to large-cluster dissipation (Jasplakinolide, Supplementary Fig. 7E) or cluster formation (Blebbistatin, Supplementary Fig. 7F). Therefore, we decided to further investigate the role of myosin-II positioning and the contractile outcome in the evolution of cortical spectrin clusters. To visualize myosins, we transfected fibroblasts of the GFP-tagged myosin-IIA (MIIA) and performed TIRFM investigations. At the cortical level, a clear complementarity between spectrin clusters and GFP-MIIA was observed (Supplementary Fig. 9A). As expected, MIIA positioning in fixed samples largely matched the F-Actin staining in cells fully adherent to the substrate or confined by the linear fibronectin pattern previously implemented (Fig. 4A).

To better understand the spatiotemporal evolution of the actomyosin-spectrin dichotomy, live TIRFM imaging was performed in fibroblasts transfected for GFP-βII-spectrin and mCherry-myosin light chain (MLC), to unbiasedly label all non-muscle myosin-II isoforms. To emphasize different myosin dynamics depending on the different cortical landscapes, fibroblasts were seeded on linear patterns to create controlled discontinuity in cortical territories, either enriched or depleted of stress fibers and adhesions. This strategy allowed us to induce cortical zones of different actin-spectrin densities and myosin-II dynamics (Fig. 7A and Supplementary Movie 6). At the single cell level, we observed regions with medium, high, and low levels of MLC puncta corresponding to medium, low, and high spectrin densities (Fig. 7A, zooms 1-2-3 and Supplementary Movie 6). In correspondence with the high-density spectrin zone, a higher and more persistent correlation between GFP-βII-spectrin and MLC puncta was observed than at stress fibers enriched regions (Supplementary Fig. 9B–E). This observation highlighted again the complementarity between actomyosin and the spectrin meshwork, but still lacked dynamic insights. MLC particle tracking was performed and highlighted the appearance of cortical MLC puncta as well as puncta associated with stress fibers (Fig. 7B, C and Supplementary Movie 6). When the lifetime of puncta was more carefully analyzed, we observed long-lived MLC tracks to be preferentially associated with the stress fibers (track length >150 s, Fig. 7B, E). Instead, short-lived MLC pulses stochastically occurred also in correspondence with denser GFP-βII-spectrin signal (Fig. 7A zoom 3

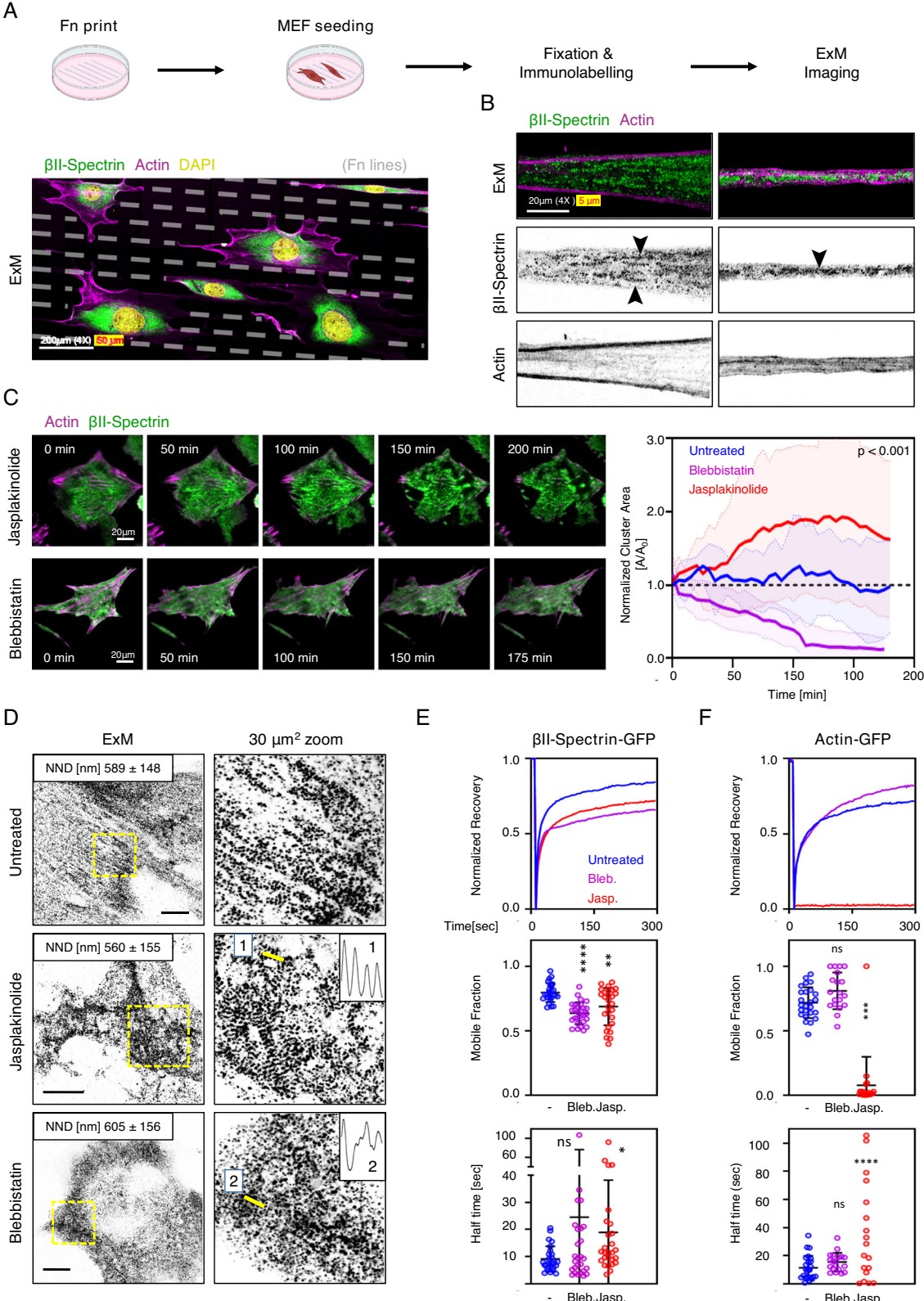

and and Fig. 7C, E), suggesting a more pulsatile MLC dynamics occurring in cortical regions characterized by prominent spectrin enrichment. These stochastic cortical myosin-II pulses have been observed to be dependent on the local recruitment and activation of contractility rather than the long-range diffusion of pre-assembled myosin rods[44]. Indeed, increased signal correlation between GFP-βII-spectrin and MLC during Blebbistatin washout was observed, as well as

an increased overlap coefficient between high-intensity spectrin clusters and MLC puncta (Supplementary Fig. 9F, G). Those pulses influenced spectrin density (kymograph, Fig. 7D), but mostly transiently as they are unstable and can dissipate rapidly. Altogether these results showed that transient local association of myosin-II at regions characterized by the absence of stress fibers drive spectrin motion and influence its local clustering.

**Fig. 4 | Jasplakinolide treatment drives spectrin clusters formation.**
**A** Schematic pipeline and representative ExM image of MEFs seeded on microfabricated adhesive lines (4 µm adhesive cross-section, 12 µm non-adhesive surface) and immunolabelled for endogenous βII-spectrin (green) and β-actin (magenta). A large volumetric imaging was performed by tile scan approach, and a projection of multiple planes is shown (scale bar: 200 µm). **B** Two protrusions from independent cells are shown to highlight the differential positioning βII-spectrin and β-actin, black arrowheads indicate clusters with incomplete periodic organization (scale bar: 20 µm). Images are representative of at least 3 independent experiments. **C** Live imaging by TIRF microscopy of MEFs transiently transfected with GFP-βII-spectrin (green) and RFP-actin (magenta), treated with Jasplakinolide 100 nM and Blebbistatin 10 µM for 3-4 h. Relevant frames are shown to highlight the differential effects on cell shape and protein clustering the two drugs induce (scale bare 20 µm). Cluster area normalized to the initial frames is calculated in response to each treatment and plotted over time ($n = 10$ independent cells, data are presented as mean ± SD, statistical analysis: one-way ANOVA of values at $t = 100$ min). **D** ExM images of MEFs treated for 3-4 h with 10 µM Blebbistatin and 100 nM

Jasplakinolide, immunolabelled for βII-spectrin (scale bars: 20 µm). 30 µm² zooms are shown, corresponding to the yellow boxes, to highlight the differential effects of the drugs on spectrin organization. NND values are reported, as well as line scans related to the yellow lines in the zooms (1 and 2). Images are representative of at least 3 independent experiments. **E, F** Recovery curves resulting from the FRAP assay are shown for GFP-βII-spectrin and GFP-Actin, transiently transfected in MEFs and treated 3-4 h with the different cytoskeletal impairing drugs (Jasplakinolide 100 nM, Blebbistatin 10 µM). Mobile fractions are reported in the graphs (data are presented as mean ± SD, statistical analysis one-way ANOVA with multiple comparisons, \*\*\**p*-value < 0.005, \*\*\*\**p*-value < 0.0001, n = 25-28(GFP-βII-spectrin) and 19-25(RFP-actin) cells in 3 independent experiments). Half-time recovery rates resulting from the fitting of the raw data with a one-exponential equation are reported in the graphs (data are presented as mean ± SD, statistical analysis one-way ANOVA with multiple comparisons, \**p*-value < 0.05, \*\*\*\* *p-value* < 0.0001, $n = 25$-28(GFP-βII-spectrin) and 19-25(RFP-actin) cells in 3 independent experiments). Panel (**A**) was created with BioRender.com and released under a Creative Commons Attribution-NonCommercial-NoDerivs 4.0 International license.

## The inclusion of myosin in the network model results in a more periodic configuration with lower stresses

To examine how myosin affects the spectrin topologies and in agreement with our experimental observations, two different types of myosin dynamics were introduced in the network model: stiff myosin linkers that connect the spectrin mesh to the stress fibers and myosin rods within the spectrin network (Fig. 7F and Supplementary Movie 7), corresponding to the long- and short-lived MLC puncta. Both types of myosin, linkers, and rods, are represented by edges with nodes at their ends. Myosin edges are contractile, and we assumed that myosin linkers are stiffer than myosin rods (see Supplementary Table 4 for modeling parameters). As we never observed clear colocalization between cortical MLC puncta and spectrin, myosin nodes were placed in the center of the triangles formed by spectrin bundles, and when one of these bundles detaches, the corresponding myosin node also detaches and reattaches to another free spectrin triangle. If there are no free spectrin triangles within a range, then the myosin is removed from the network. Simulations show that the inclusion of myosin alters the final configuration of the network, lowers the stresses, and changes the spatial configuration of the number of spectrin bundles connected by a short actin filament (Supplementary Fig. 10A, C, E). Moreover, the final distribution of spectrin length is close to the resting length (Supplementary Fig. 10F). We then asked what happens when the myosin rods are added and removed stochastically. In this case, the resulting configuration of the spectrin mesh was improved by reducing the number of spectrin bundles attached to the short actin filaments at the center of the network, thus better resembling the periodic configuration (Fig. 7F, G and Supplementary Fig. 10B, D). Furthermore, the model with stochastic myosin rods was able to replicate experimental observations, such as the low number of myosins in the network over time (Supplementary Fig. 10G) and that the dynamic myosin has a shorter lifetime than the myosin attached to the stress fibers, which remains throughout the simulation (Fig. 7E, H and Supplementary Movie 7). Note that the contraction generated by myosin allows the relaxation of the network in zones that were stressed in the previous model (compare Supplementary Fig. 10A, B with Fig. 6I). The interaction between spectrin, myosin, and stress fibers must be balanced to obtain physiological possible configurations (Supplementary Fig. 10H–M). Taken together, our computational model shows that stochastic myosin effectively reduces the stress in the spectrin network and improves its transition to a periodic, clustered configuration.

## Discussion

Dynamic changes in cytoskeletal scaffolds are emerging as fundamental mechanisms to convert cell shape adaptation by mechanical cues (mechanoadaptation) into specific activation of transduction pathways (mechanoresponse). Membrane-attached cortical elements represent the primary responsive material that cells can deploy to sense and react to these external perturbations but are inserted in a crowded milieu notoriously difficult to disentangle by optical and electron microscopy[45,46]. Here, we identified the periodic topological organizations of the spectrin meshwork in fibroblasts and investigated the contribution of the local mechanical landscape of the cell cortex in driving this topological transition. Experimental and literature-derived parameters (see "Methods") allowed us to build a computational model that integrates actomyosin dynamics, mechanical forces, and spectrin turnover and predicted the molecular events required for the transition toward this high-density spectrin conformation. In particular, spectrin detachment as a consequence of reduced mechanical stress represents the necessary geometrical requirement to adopt the periodic meshwork organization; the transition is otherwise impossible by applying the sole mechanical stress on the stereotypical isotropic spectrin lattice. If correct, this model describes spectrin's capability to respond to local mechanical fluctuations mainly induced by pulsatile actomyosin contractility and to create spectrin reservoirs in the form of periodic clusters. However, this in silico observation, which represents an intriguing and elegant stress-adaptation mechanism, requires further experimental evidence that is beyond our technical possibilities.

At the whole-cell scale, a functional role for the asymmetric distribution of spectrin has started to emerge during neuronal cell migration and dendrite development in *C. elegans*[47]. The preferential posterior distribution of spectrin and the predominant Arp2/3 activity at spectrin-depleted leading edges suggest how spectrin remodeling influences local cortex mechanics by preventing ectopic actin assembly in migrating neural progenitors. We observed a similar exclusion of spectrin from the lamellipodia of spreading fibroblasts, as well as de novo polymerization of branched actin following the displacement of spectrin induced by compressive stimuli[14]. Here, we provide unprecedented details on the heterogeneous distribution of the spectrin meshwork that can be observed within the same cell cortex. During epithelial morphogenesis, asymmetric spectrin distribution is also critical[48]. Indeed, the correct integration of cell shape regulation, cortical mechanics, and Hippo signaling effectors is disrupted in the spectrin-deficient Drosophila retina, which lacks the apical spectrin accumulation necessary to correctly develop the regular array of hexagonal ommatidia[49,50]. Moreover, dual spectrin/plastin inhibition results in cytokinesis failure due to disorganization and collapse of the equatorial actomyosin network in *C. elegans*[51]. Altogether, these observations point to the capacity of spectrin to create dense buffering reservoirs in cortical regions characterized by low mechanical

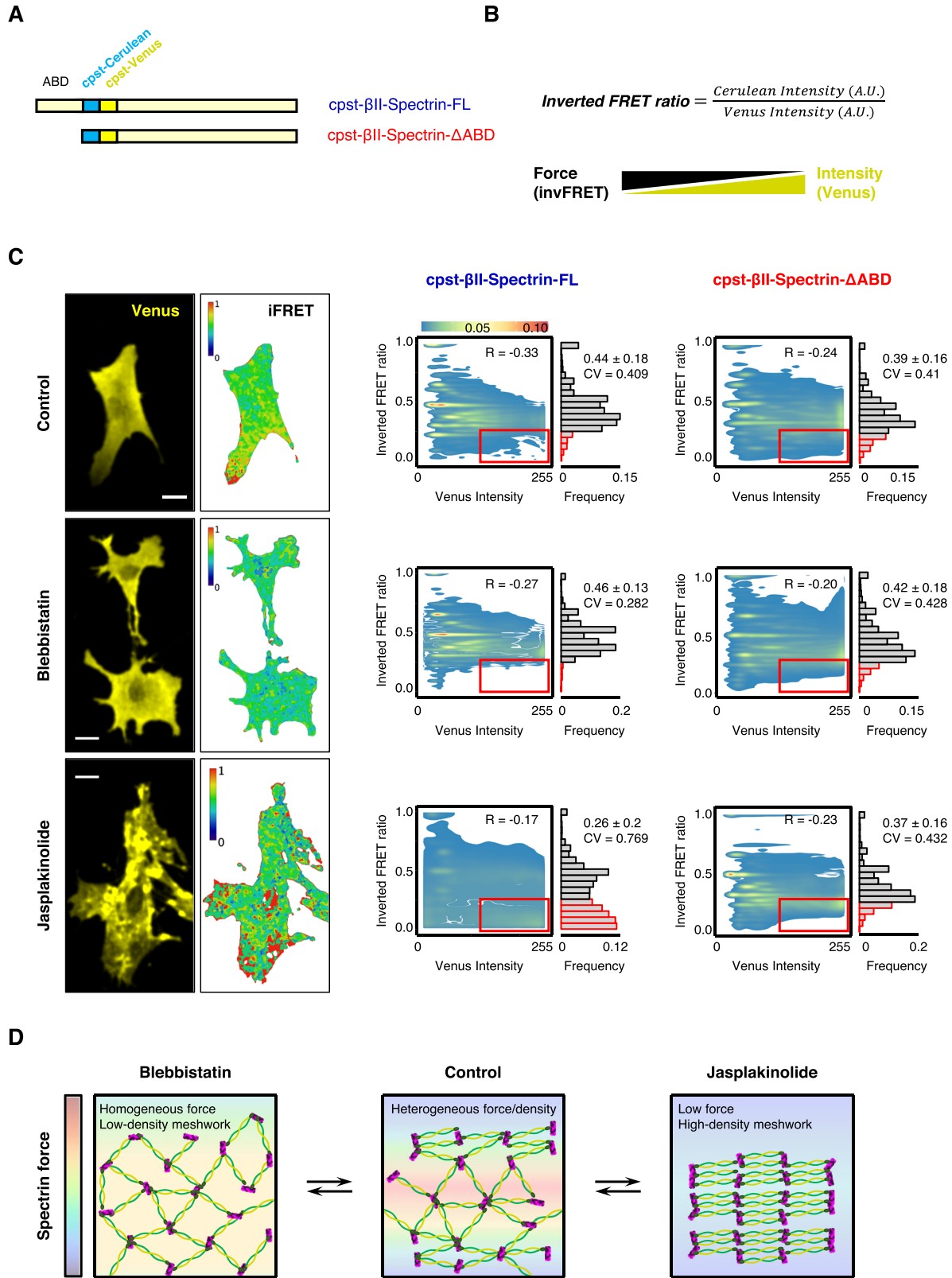

stress, by adopting the ultimately packed periodic topology. Upon changes in cortical mechanics, these reservoirs can be deployed to secure a continuous, albeit heterogeneous, coverage of the plasma membrane.

Historically, the spectrin scaffold is considered a skeletal integrator[52], capable of organizing the logistic distribution of a plethora of binding partners and cytoskeletal and membrane trafficking

events. We reported reduced AP2-dependent endocytic events in correspondence with dense spectrin regions induced by osmotic fluctuations[14]; similarly, in neurons, clathrin "packets" associate with the spectrin periodic scaffold along the axons but never fused to the adjacent plasma membrane, suggesting the existence of a membrane fusion barrier in presence of a dense spectrin array[23,53]. Recent proteomic investigations identified proteins specifically associated with

**Fig. 5 | Spectrin displays a low tensional state in clusters. A** Schematic representation of the βII-spectrin FRET-based tension sensor. ΔABD (red) lacks the actin-binding domain, while the FRET pair (cpst-Cerulean and cpst-Venus) is inserted between the Actin Binding Domain and the central rod in the full-length counterpart (FL, blue). **B** InvFRET is calculated according to Meng and Sachs (2012): higher ratio reflects a higher tensional state (lower FRET) and vice versa, the expected relationship between tension and fluorescence intensity of the acceptor is also reported. **C** Representative FRET images are shown (scale bar 20 μm). Pixel-by-pixel analysis was performed in at least $n$ = 90 cells per condition, and scatter plots show the local relationship between the invFRET and Venus intensity. Contours of 2D

Kernel densities are reported with the corresponding calibration bar, Pearson's correlation coefficients as well as coefficient of Variation (CV, defined as the ratio of the standard deviation to the mean) are also reported. The frequency distribution of the invFRET highlights the peculiar behavior of the FL βII-spectrin FRET-based tension sensor upon Jasplakinolide treatment, which displayed a unique bimodal distribution with the appearance of low invFRET values in correspondence of high-intensity Venus (highlighted by the red box). The same was not observed in any other experimental conditions. Mean ± SD values are reported. **D** Cartoon representation of the differential configurations of the spectrin mesh upon drug perturbations, and the relationship with tension.

the periodic scaffold and not with the extended meshwork[54]: in particular the structural WD-repeat containing proteins (i.e., coronins), transmembrane channels and receptors (i.e., potassium and sodium ion channels, GPCRs, RTKs), and the signaling molecule CAMK IIβ. Our observations suggest spectrin clustering as a driver for transmembrane and membrane-associated protein organization also in different cell lineages where spectrin is highly expressed and potentially explains the integrative role of shear forces and mechanoresponse recently reported in endothelial cells[55–57], with immediate consequences on cell physiology in homeostasis and disease.

The parameters provided here lend themselves to resolving important questions regarding neuronal development. Indeed, the way in which spectrin reaches the stereotypical 3D barrel-like periodic organization in axons represents an informative and specific developmental mechanism. We identified key events required to transition from the non-polarized to a flat periodic spectrin organization. In particular, actomyosin contractility and the constraints imposed by the actin stress fibers are the main drivers of these topological rearrangements. Knowing that membrane mechanics can also drive spectrin reorganization[14], we can reasonably integrate these observations and hypothesize that developing neurons can pack spectrin into the mature barrel with the additional help of constraints imposed by axonal membrane tubulation. This is suggested by our observations in fibroblasts adhered on printed lines, and it would be interesting to study how this topological spectrin transition can be directed in differentiating neurons seeded on these instructive surfaces.

## Limitations to the study
Dynamic analyses of protein behaviors were performed using transient overexpression of fluorescently-tagged exogenous proteins (i.e., GFP-βII-spectrin, cpst-βII-spectrin, RFP-actin, mCherry-MLC, etc.). The aim of these standard cell biology approaches was to confirm the observations made endogenously by TIRFM or ExM. While we obtained similar results on transiently expressing *versus* endogenously stained cells subjected to various manipulations, we cannot exclude that some of the results may be biased by the excess of exogenous protein. To mitigate this bias, we only examined cells with low-to-moderate transfection efficiency. Future work should focus on improving the resolution of live experiments to directly address the transition of spectrin between the topological organizations described here (periodic *versus* diffused), and their relative frequency at the cell scale. A triangular mesh was chosen as the starting configuration in our model, despite experimental evidence suggesting a diffused organization in fibroblasts. This idealized configuration allows for uniform tiling of the 2D space and hence, avoids initial clustering and allows us to examine conformational transitions. Due to the stochasticity of our model, quantifying changes in clustering from a random initial configuration could lead to ambiguity in the model outcome.

## Methods
Our research complies with all relevant ethical regulations and is approved by IFOM ETS and the University of California San Diego.

## Cell culture
Immortalized MEFs derived from RPTP α + /+ murine background[58] were grown in complete media composed of DMEM (Lonza) supplemented with 10% Fetal Bovine Serum (FBS South America, Euroclone) and 2 mM L-glutamine at 37 °C and 5% CO2. For imaging experiments, MEFs were seeded on borosilicate glass coverslips of 1½ thickness (Corning) or Nunc Glass Base Dishes (Thermo Fischer Scientific) coated with sterile 10 μg ml$^{-1}$ fibronectin (Roche). The supplier and identifier for all reagents are listed in Supplementary Table 5. Cytoskeletal drug perturbations were performed by supplementing complete media with 10 μM Blebbistatin and 100 nM Jasplakinolide (Merck) for 3-4 h. Other drugs were tested at different concentrations for the same incubation time, specifically: CK666 (100 μM), SMIFH2 (20 μM), Latrunculin A (1 μM), Cytochalasin B (1 μM), and Cucurbitacin E (5 nM). Transient expression of the plasmids listed in Supplementary Table 5 was achieved by Neon Transfection System Microporator (Thermo Fischer). Briefly, $1 \times 10^6$ cells were trypsinized, washed once with PBS, and mixed with a total of 10 μg of recombinant DNA in electroporation buffer R (Invitrogen). Cells were singularly electroshocked at 1600 mV for 20 ms by placing the electroporation tip into the column filled by E2 buffer (following the manufacturer's specifications). Cells were seeded and allowed to recover for 24 h on plastic dishes or glass coverslips.

Genome-edited MEF cell lines were generated by CRISPR-Cas9 technology. Specifically, cells were transiently transfected with the plasmid pSpCas9(BB)−2A-GFP (PX458) (Addgene) encoding Cas9 and GFP reporter for positive transfection. After 24 h, cells expressing Cas9 were electroporated with 1 nmol of synthetic single-guide RNA directed against exon 3 of the murine gene *sptbn1* (Invitrogen). With the help of IFOM cell sorting and cell culture facilities, after 48 h from the initial transfection GFP positive cells were sorted by arbitrarily choosing an intermediate level of fluorescent intensity and cultured at clonal dilution in 96-well plates. Clonal populations were expanded and successful KO screened by western blots for protein expression. Among the 50 clones screened, we selected clones 8, 9, and 10 as *sptbn1* KO and clone 15 as *sptbn1*+/+ control for Cas9 off-target effects.

## Erythrocytes and hiPSC-derived cortical neurons
Erythrocytes were obtained from 6-12 weeks old C57/J male mice (Charles River), immediately after euthanasia in accordance with the guidelines established in the Principles of Laboratory Animal Care (directive 86/609/EEC) and were approved by the Italian Ministry of Health. Erythrocytes were washed in PBS supplemented with 10 mg ml$^{-1}$ glucose, harvested by centrifugation at $300 \times g$ for 5', resuspended at the desired concentration, immobilized, and fixed with 4% PFA in PBS on PLL-coated glass coverslips for immunofluorescence investigations as reported[27].

hiPSC-derived cortical neurons were generated by the IFOM Cell Culture facility following the NGN2-inducible system described in ref. 59. Briefly, NGN2 hiPSC cells were cultured on Matrigel-coated glass coverslips in Induction Media (DMEM, 2 mM L-glutamine, 10 mM NEAA) supplemented with 2 μg ml$^{-1}$ Doxycycline and 10 μM Y27632. On day 1, after seeding, Y27632 was withdrawn from the media. At day 3, hiPSC were detached and seeded again on Matrigel-coated coverslips

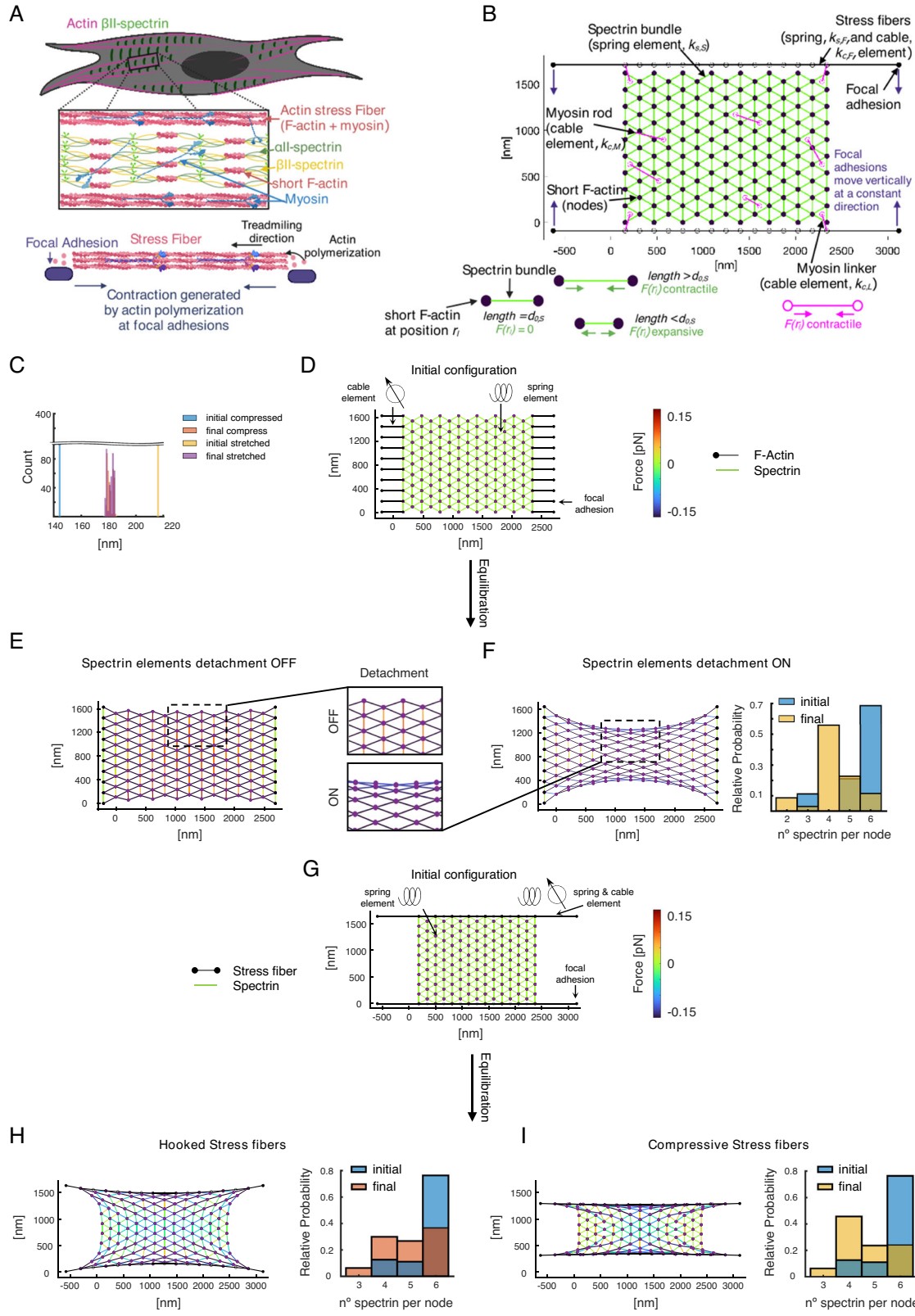

in Cortical Neuron Culture Media (Neurobasal (Thermo Fischer), 2 mM L-glutamine, 10 ng ml⁻¹ NT-3, 10 ng ml⁻¹ BDNF, 10 µg ml⁻¹ laminin) containing 2 µg ml⁻¹ Doxycycline and cultured up to 21 days by exchanging the media every 48 h. At the desired time points, coverslips were washed in PBS and fixed with 4% PFA for immuno-fluorescence investigations.

## Micropatterning

Borosilicate glass coverslips (Corning) were washed for 1 h with 20% Acetic Acid, rinsed in Milli-Q water, and stored in 90% ethanol. When needed, coverslips were air-dried and activated by a plasma cleaner (Harrick Plasma) for at least 3 min. The surface was passivated by incubation with 0.1 mg ml⁻¹ poly(ethylene glycol)-b-poly(l-lysine)

**Fig. 6 | Theoretical model of a spectrin patch. A** Idealized cell showing periodic spectrin (green bundles) between stress fibers (magenta). The square box shows a magnification of the spectrin cluster. A schematic picture of the stress fibers (pink) attached to the extracellular space through focal adhesions (purple) creating contractile stress is shown at the bottom. **B** Initial configuration of the modeled spectrin cluster showing spectrin bundles (green edges), myosin (magenta), and stress fibers (black lines). The short actin filaments are depicted by purple circles. Black full circles correspond to focal adhesions that can move vertically, as shown by the purple arrows. **C** Histogram showing the initial and final length of spectrin filaments that are initially smaller (larger) than the resting length thereby acting as a compressed (stretched) spring and exerting a restorative positive (negative) force on the mesh. The time to reach such a state is 60 s. The corresponding meshes are in Supplementary Fig. 8F, G. **D** Spectrin mesh with spring elements attached to fixed focal adhesions (black dots) through cable elements (black edges). **E** Final

configuration after equilibration (120 s) of the network without allowing for any spectrin removal for the mesh shown in (**D**). **F** Configuration of the network in (**D**) after equilibration allowing spectrin bundles to detach when the force generated by the spring element is above $F_{th}$. The histogram shows the number of spectrin bundles per short-actin filament. Zooms of the final configurations are highlighted in squared boxes. **G** Spectrin mesh with the spring elements attached to stress fibers represented by black edges. The stress fibers have spring and cable elements connected by black empty circles. Solid circles represent fixed focal adhesions. **H, I** Final configuration of the network in (**G**), allowing spectrin bundle removal. In (**I**), focal adhesions are vertically pulled together during the first half of the simulation. In the second half, focal adhesions are fixed. The total duration is 600 s. Histograms represent the number of connected spectrin bundles per short-actin filament.

(PEG–PLL, Ruixibio) for 1 h at room temperature to prevent fibronectin coating. A quartz mask (Delta mask B.V.) was washed with isopropanol and activated under UV light for 7 min (UVO Cleaner, Jelight). PEGylated coverslips were aligned to the desired pattern in the mask, and illuminated under UV light for 7 min. The quartz layer prevents UV illumination of the passivated surface while the photolithography-made pattern allows the light to pass, burning the PEG–PLL. Patterned coverslips were then coated with 10 µg ml⁻¹ fibronectin for 1 h at room temperature, while the passivated surface prevented fibronectin adherence. After rinsing the coverslips several times with PBS, MEFs were seeded at the desired cell density and cultured at 37 °C in the same media described before.

### Membrane fragility assay
96-well black bottom plates were coated with 10 µg ml⁻¹ fibronectin, cells seeded at $2 \times 10^3$ cells per well density and left undisturbed for 24 h. Membrane fragility was evaluated by performing CellTox™ Green cytotoxicity assay (Promega) under different osmotic conditions in Ringer's buffer (see Supplementary Table 5 for buffer composition). This assay records the increase in fluorescence of the dye that can penetrate the cell and bind to the DNA only in the presence of a damaged plasma membrane. Otherwise, the dye remains extracellular and at low fluorescence emission. Before the beginning of the experiment, all wells and cells were equilibrated in Ringer's 1x for 30 min. A replica 96-well plate was prepared by dissolving CellTox™ Green dye 1:1000 in Ringer's 1x, 1.5x, and 0.5x solutions, including positive control conditions where media was supplemented with 30 µg ml⁻¹ digitonin, and negative controls to record no-cell background. Immediately before the experiment, all the wells in the experimental plate were emptied and filled with the respective ones of the replica plate with the help of a multichannel pipette to limit the delay between the beginning of the osmotic shocks and the recording of fluorescence intensities. Fluorescence was measured at 485–500 nm$_{Ex}$/520–530 nm$_{Em}$ using a 96-well fluorescence plate reader equilibrated at 37 °C, recording fluorescence reads every 2 min for 3 h. Each experimental condition was recorded in triplicate, experiments were qualitatively assessed and discarded when positive controls treated with digitonin failed to show an immediate steep increase in fluorescence before reaching a plateau. The average of the three wells per condition was calculated and normalized to the initial time point. Values were analyzed and plotted with the software GraphPad Prism. Given the noisy nature of the single record, to avoid over-estimating or under-estimating plateau values at given time points, fluorescence intensity curves were fitted to the one-phase association equation to obtain estimates of the plateau fluorescence for each experimental condition.

### Immunofluorescence
The antibodies used in this study were the following: mouse anti-βII-spectrin (dilution 1:200, BD-bioscience), rabbit anti-βII-spectrin (1:200,

Abcam), mouse anti-βI-spectrin (dilution 1:200, NeuroMab), mouse anti-αII-spectrin (1:200, Invitrogen), and rabbit anti-β-actin (1:100, Cell Signaling), rabbit anti-adducin (1:100, Abcam). Before fixation, cells were seeded on 10 µg ml⁻¹ fibronectin-coated coverslips/glass base dishes. Cells were fixed in 4% paraformaldehyde for 10 min, then neutralized using 10 mM NH₄Cl in PBS for 10 min. Alternatively, fixation was performed in ice-cold pure methanol for 2 min at − 20 °C. Cells were subsequently washed three times with PBS, permeabilized for 2–5 min using PBS containing 0.1% Triton X-100, and blocked with 3–5% BSA for 10 min at room temperature. Cells were incubated with primary antibody overnight at 4 °C. After 3 washing steps in PBS, cells were incubated with CF568/AttoN647-conjugated goat anti-mouse or anti-rabbit (1:100–1:400, Thermo Fischer Scientific) and Alexa 488/568-conjugated phalloidin (1:200, Sigma-Aldrich) for 1 h at room temperature. After three washes in PBS, cells were mounted with anti-fade glycerol-based media (for confocal microscopy) or PBS (for TIRFM and ExM) and stored at 4 °C. All primary antibodies and fluorophore-conjugated secondary antibodies are listed in Supplementary Table 5.

### Expansion microscopy (ExM)
The procedure for ExM was adapted from the original report[26]. For completeness of information and reproducibility all sensible steps are described. The immunofluorescence procedure was performed as described, except for the fluorophores used: CF568-conjugated goat anti-mouse (1:100 Sigma-Aldrich) and AttoN647-conjugated goat anti-rabbit (1:100, Sigma-Aldrich). When ExM was performed on micro-patterned coverslips before the immunostaining procedure, a quenching step of 2 min at − 20° with ice-cold methanol was required to prevent reactivity of the PEG-PLL surface coating with the amino-reactive Anchoring buffer.

### Anchoring
Upon completion of the immunostaining procedure, the specimen on coverslips were incubated with the Anchoring buffer, consisting of PBS supplemented with 1 mM of the amino-reactive MA-NHS (Methacrylic Acid N-Hydroxysuccinimide Ester, Sigma-Aldrich). Specimens were incubated for 1-1.5 h on gentle rocking at room temperature.

### Gelation
The gelation solution was composed of the Monomer stock (sodium acrylate 33 wt%, acrylamide 50 wt%, bis-acrylamide 1 wt%, 1.8 M NaCl, 1x PBS, Sigma-Aldrich), supplemented before gelation with 0.2 wt% of TEMED and Ammonium Persulfate (APS, Sigma-Aldrich). The gelation solution was supplemented with 0.01 wt% of 4-Hydroxy-TEMPO (Sigma-Aldrich) to slow down the reaction and allow complete diffusion inside the cells. Anchoring buffer and excess of MA-NHS was removed by 2x brief washes with PBS. Gelation was performed in a custom-made gelation chamber: 2 microscopy slides were coated with

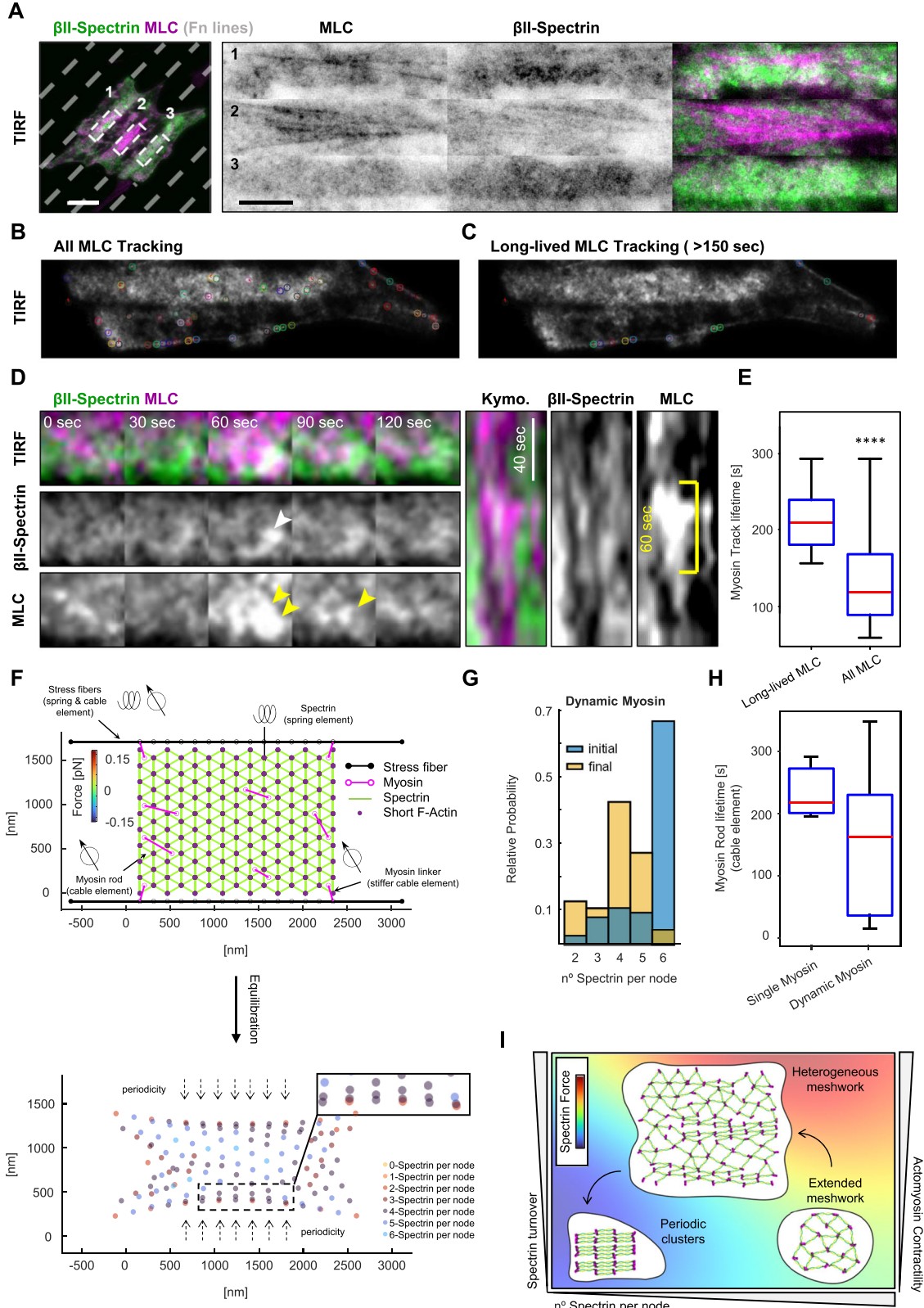

parafilm and spaced by silicone isolator gaskets of 0.5 mm thickness. Coverslips were placed on the bottom side of the chambers, biological specimens facing the inner side of the chamber, and excess of PBS buffer was removed. A Complete Gelation Solution was added and chambers were placed flat on ice for 30 min to favor homogeneous diffusion of acrylamide and avoid gelation artefacts. After this step, chambers were placed for 1 h at 37°.

## Digestion

Gelation chambers were carefully dismounted by removing the top slide, and the gel was trimmed with a razor blade. Gel shapes with no-mirror symmetry are useful to recognize the surface of the gel with the biological specimens located at the interface. Coverslips with trimmed gels still attached were placed in an opportune plastic dish and incubated with Digestion Buffer (50 mM Tris-HCl, 125 mM NaCl, 2.5 mM

**Fig. 7 | Myosin II dynamics act as the dominant force for spectrin clustering.**
**A** Live imaging by TIRF microscopy of MEFs transiently transfected with GFP-βII-spectrin (green) and mCherry-MLC (magenta) seeded on microfabricated adhesive lines (not stained, gray dashes) to force discrete cortical organizations (scale bar = 20 μm). Live images are representative of at least 3 independent experiments. Three different regions are highlighted and display representative cortical dynamics (1–3). Automated tracking of mCherry-MLC puncta with two different filtering parameters: in (**B**), all tracks are shown in a single representative frame, while in (**C**) only long-lived tracks (>150 s) preferentially localized on stress fibers. **D** Pulsative myosin dynamics is shown at spectrin-rich cortical domains (5 × 5 μm zoom). GFP-βII-spectrin and mCherry-MLC clustering are highlighted by white and yellow arrowheads respectively. Kymograph qualitative analysis showed fenestration in the GFP-βII-spectrin signal (white arrowheads) in correspondence with the mCherry-MLC pulse (yellow arrowhead). The yellow bar highlighted an MLC pulse of ≈ 60 s. **E** Differential lifetime of MLC puncta presented in (**B**) (All MLC Tracking) and (**C**) (Long-lived MLC Tracking), data are presented as the center (median), box (25th to 75th IQR), whiskers (min and max) (*n* = 2200-6965 tracks in 5 independent cells, statistical analysis Mann-Whitney test ****p*-value < 0.0001). **F** Top: initial configuration of the theoretical model where the spectrin mesh is attached to the stress fibers (black edges) through myosin linkers with cable elements (magenta lines, with empty circles at their ends). Myosin rods with less rigid cable elements are randomly distributed in the mesh. Focal adhesions are pulled together at a constant velocity during the first 300 s of the simulation, after which, they are held fixed. The total duration of the simulation is 600 s. Bottom: position of the short-actin filaments at the end of the simulation (equilibration), color-coded to represent the number of attached spectrin bundles per node. **G** Histograms denote the initial and final number of spectrin bundles per short-actin filament. **H** Boxplot of the lifetime of the myosin rods in the spectrin network for the two different simulation conditions (*n* = 5 single myosin, *n* = 11 dynamic myosin), data are presented as center (median), box (25th to 75th IQR), whiskers (min and max). **I** The phase diagram integrates the different experimental and theoretical parameters identified in this study to explain the contribution of the different players in the topological transition of the spectrin meshwork. Panel (I) was created with BioRender.com and released under a Creative Commons Attribution-NonCommercial-NoDerivs 4.0 International license.

EDTA, 0.5% Triton X-100) supplemented with Proteinase K (1:100, New England Biolabs). Specimens were incubated on an orbital shaker at 37 °C per 2 h and 60 rpm. hiPSC-derived cortical neurons cultivated on Matrigel were digested at 30 °C Over Night and 60 rpm orbital shaking.

## Expansion
For the physical expansion step, gels were transferred into bigger plastic dishes to ensure undisturbed planar expansion. The digestion buffer was replaced by Milli-Q water and incubated with gentle shacking. At least 4 washing steps of 30 min were performed. Expanded gels were kept at 4 °C in the dark.

## Mounting
Acrylamide hydrogels required immobilization on glass coverslips to avoid drift during volumetric imaging. Coverslips were plasma activated (Harrick Plasma) for 3 min and then coated with poly-L-Lysine for 1 h at room temperature, washed in Milli-Q water, and dried before letting the gel settle.

## TIRFM and confocal microscopy
Confocal microscopy was performed on a Leica TCS SP8 laser-scanning confocal module mounted on a Leica DMi 8 inverted microscope, equipped with a motorized stage, and controlled by the software Leica Application Suite X (ver. 3.5.2.18963). For image acquisition, a HC PL APO CS2 63 × /1.40 oil immersion objective was used. DIC, epifluorescence (EPI), and total internal reflection fluorescence microscopy (TIRFM) of fixed specimens, live time lapses, and drug treatments were performed on a Leica AM TIRF MC system. Two different TIRFM-grade objectives were used: HCX PL APO 63 × /1.47NA oil immersion and HCX PL APO 100 × /1.47NA oil immersion. Three different laser lines were used for fluorochrome excitation: 488 nm, 561 nm, and 635 nm. A specific dichroic and emission filter set for each wavelength has been used. The microscope was controlled by Leica Application Suite AF software (ver. 2.6.1.7314), and images were acquired with an Andor iXon DU-8285_VP camera and 100−400 msec exposure times depending on signal intensity. For live imaging experiments, environmental conditions were maintained by an Okolab temperature and $CO_2$ control system.

## STED microscopy
STED super-resolution microscopy was performed on a Leica TCS SP8 laser-scanning confocal module mounted on a Leica DMi 8 inverted microscope, equipped with a 775 nm pulsed depletion laser, and controlled by the software Leica Application Suite X (ver. 3.5.2.18963). For image acquisition, a HC PL APO CS2 100 × /1.40 oil immersion STED WHITE objective was used. Biological samples were immunolabelled with the STED compatible secondary antibodies conjugated with Atto-594 and Atto-647N (Sigma-Aldrich), while F-actin was labeled with Phalloidin-Abberior STAR 635 (Merck). Samples were mounted in Mowiol anti-fade. Imaging was performed with a tunable pulsed White Light Laser set to the two excitation wavelengths of the fluorophores, and signal depletion was performed with a 770 nm pulsed depletion laser at 50−60% output power. Images were deconvolved by the software Huygens Professional, considering a theoretical PSF.

## Transferrin uptake and cell area analysis
Clonal cell lines were seeded on fibronectin-coated glass-bottom 24-well plates at the density of $1 × 10^4$ cells per well and let them undisturbed for 24 h. Cells were serum-starved in DMEM for 30 min before supplementing them with DMEM, 10% FBS, 2 mM L-glutamine, and 50 μg ml$^{-1}$ of rhodamine-labeled transferrin (Invitrogen). At different time points (0, 10, 20, and 30 min), excess of transferrin from the cell surface were stripped by a fast wash in 10x PBS, and fixed in 4% PFA. Cells were incubated in PBS containing 0.1% Triton X-100, 3–5% BSA, 1:100 Alexa 488-Phalloidin, and 1:5000 DAPI for 30 min at room temperature. After three washing steps of 5 min in PBS, imaging was performed on a Leica DMi8 Thunder Widefield microscope equipped with HC PL APO 40 × /0.95NA air objective, motorized stage, and camera at constant settings. Image analysis was performed in Fiji by creating two independent binary masks of the DAPI and phalloidin channels. Particles adjacent to the image border or that did not contain nuclei were excluded. Clusters of cells were manually segmented when possible and a final Analyze Particle tool was run in Fiji to segment objects comprised between 300 and 20000 μm$^2$. Individual object area values were exported to GraphPad Prism for final analysis. Mean rhodamine-labeled transferrin intensity signals were obtained by first subtracting the background with the sliding paraboloid method in Fiji (rolling ball radius 50 pixels) and lastly by applying a Gaussian blur filter (radius 2 pixels). The image was multiplied by the binary mask generated to extract cell area values, and the mean rhodamine intensity for each individual cell was calculated. Values were exported to GraphPad Prism for statistical analysis and graphical representation.

## Fluorescence intensity distribution
The TIRFM dataset was analyzed to extract Intensity distribution in Fiji. Briefly, images were acquired at constant laser intensity, exposure time and no electronic gain between independent channels. Images were converted to 8-bit format and were down-sampled to the pixel size of 1 × 1- μm by interpolating the bilinear average. A binary cell mask was generated by the "Analyze Particle" tool, and a cell outline was added to the ROI manager. For each channel, the Histogram of intensity

distribution was generated, and the count value matched the projected cell area in µm2. Extracted values were normalized by the total count to obtain a relative frequency and allow averaging between multiple cells.

## Clusters segmentation

TIRFM images were processed by the tool "Subtract Background" in Fiji, and the method Sliding paraboloid with a rolling ball radius of 50.0 pixels was applied. Single-channel images of the 16-bit format were independently thresholded with the default method by imposing a cutoff value of 5%. This approach sets the 95% of the intensity distribution curve to 0, while the remaining 5% ($P_{0.95}$) is set to max. The Despeckle and Smooth tools were run to homogenize the signal, and the Analyze Particle was used to obtain Area and Shape descriptors for each individual cluster. To extract the Normalized Cluster Area from TIRFM time lapses, the same approach was applied over the entire stack. The binary mask of the clusters was obtained by applying the threshold method with a cutoff value of 5% referred to the initial frames of the time lapses, and not by calculating the 5% threshold for each individual frame. The Despeckle and Smooth tools were run, and finally, the Analyze Particle tool (Fiji) was applied to create a binary mask of objects with size 0-Infinity. The resulting mean value of the entire 8-bit binary image represents a proxy of the cluster area and can be measured over the entire stack to determine temporal variations with respect to the initial frames (an increase in the mean equals an increase in the cumulative clusters area over the entire cell).

## Orientation analysis

To analyze signal orientation on the TIRFM dataset, the Fiji plugin OrientationJ was used to quantify the local orientation properties of an image, based on the structure tensor of a defined local neighborhood[60,61]. Briefly, images were background subtracted by the Sliding paraboloid method with a rolling ball radius of 50.0 pixels. The vector field was independently calculated for each channel by applying a Gaussian gradient of 5 pixels. Coherency maps, as well as overlaid vector images, were generated. The distribution of orientations was independently calculated for each channel by the Distribution tool of the plugin. The same Gaussian gradient of 5 pixels was applied, with no Min coherency (0%) set as default. Dominant Direction was instead calculated only for the actin channel, the resulting value for the single cell under investigation was set as 0°. All the other channels were aligned according to actin dominant direction. Coherency analysis during time lapses was performed with similar parameters. The sum of the two fluorescent channels was used to create a binary mask of cells over time by the Analyze Particle tool in Fiji. ROI for each frame was added to the ROI manager and used to calculate mean coherency values for both channels over the entire stack. Values were extracted, normalized to the initial frames and plotted using the software GraphPad Prism.

## Colocalization analysis

The Fiji plugin JACoP (Just Another Co-localization Plugin) was used to determine co-localization parameters between channels, which included Pearson's correlation coefficient and Cross-Correlation coefficient with the 50-pixel shift. In particular, this function returns a series of Correlation coefficients when the two channels are shifted by ±50 pixels between each other. Costes' automatic threshold function was used to obtain the correlation coefficient in the function of the mask stringency (mCherry-MLC channel was used). Raw data were then plotted by using the software GraphPad Prism.

## Osmotic shocks and drugs washout

Assays were performed on custom-designed 2-way aluminum slides, sealed on the two planar faces by 22 × 22-mm glass coverslips welded by high vacuum grease (Sigma-Aldrich). Coverslips were acid-washed with a 20% $HNO_3$ solution before being dried and coated with 10 µg ml$^{-1}$

fibronectin. The chamber was rinsed with $CO_2$-independent 1 × Ringer's solution and equilibrated on the microscope stage at 37 °C. If required, Ringer's solution was supplemented with the drug under investigation (see Supplementary Table 5 for buffer compositions). The top and bottom glass surfaces of the chamber allow simultaneous fluorescence and DIC illumination during media exchange. MEFs were transfected 24 h before the experiment with the opportune plasmid combinations; in case of drug perturbations, cells were treated for 3-4 h before the beginning of the imaging experiments. During time lapse acquisition, the custom-designed slides allowed the exchange of media, marking the beginning of the hypo-osmotic shock or the washout of the drugs.

## FRAP experiments

MEFs expressing GFP-βII-spectrin constructs were imaged 24 h after transfection with a confocal spinning disk microscope (Olympus) equipped with iXon 897 Ultra camera (ANDOR) and a FRAP module furnished with a 405-nm laser. The environmental control was maintained with an OKOlab incubator. Images were acquired using a 100 × / 1.35Sil silicone oil immersion objective. MEFs were trypsinized and seeded on glass base dishes (Matek, Sigma-Aldrich) coated with 10 µg ml$^{-1}$ fibronectin. Before imaging, $CO_2$-independent 1 × Ringer's solution was exchanged. Circular regions of interest of 3-5 µm diameter were photobleached with the 405-nm laser at 100% intensity, and post-bleach images were acquired with 15–20% laser intensity for 100 frames (1 frame every 3 s for full-length and truncated GFP-βII-spectrin constructs and every 0.5 s for PE/ANKbs only). FRAP data were analyzed, and curves fitted to the one-exponential recovery equations (one-phase association) by the software GraphPad Prism:

$$I = I_0 + I_{max}\left[1 - e^{-kt}\right] \tag{1}$$

where I represent the relative intensity compared to the prebleach value, k is the association rate, and the half-time recovery is expressed in seconds.

## Sensitized FRET emission

**Imaging settings.** FRET recordings were performed on cells transfected with cpst-βII-spectrin-FL, or cpst-βII-spectrin-ΔABD imaged through a Leica SP8 Confocal microscope. Briefly, three channels were sequentially recorded by a HyD detector at constant electronic gain between channels: donor, FRET, and acceptor channels. Donor excitation was achieved with an Argon laser at 458 wavelengths, and spectral detection bandwidths were set at 475–485 nm; FRET excitation with the same 458 nm laser and spectral detection bandwidths 515–525 nm; acceptor excitation with 514 nm laser and spectral detection bandwidths 520–600 nm. Images were acquired with a pinhole set to 1 AU in 8-bit and 512 × 512-pixel format.

**Inverted FRET index calculation.** Raw images were processed by applying a Gaussian blur of sigma radius of 1 µm, ensuring this spatial resolution in the resulting FRET values. A binary mask of the cell is created and used to set all the values outside the cell to NaN. According to the original report[22], the inverted FRET ratio was calculated by dividing the Donor image by the FRET image. Floating values were normalized between 0 and 1. For whole-cell inverted FRET calculations, the Analyze particle tool (Fiji) was used to get the Mean gray values. Since outside the cell mask pixels were set to NaN, mean gray values only resulted from the cell projected area. For pixel-by-pixel analysis a stack of 2 channels was generated: the first channel represented the inverted FRET ratio image (mean gray values between 0 and 1), and the second channel the Acceptor image (mean gray value between 0 and 255, 8-bit format). A custom-written plugin allowed the retrieval of values in both channels for corresponding pixels, creating a.csv file to be analyzed in R. Scatter plots at each experimental condition were generated by excluding saturated pixels from the analysis.

**MLC tracking.** TIRFM time lapses of transfected MEFs were acquired at a 10 s frame rate for > 15 min. Raw images were processed in FiJi: the background was subtracted (Sliding paraboloid method with rolling ball radius of 100 pixels), bleach correction was performed by the built-in Histogram matching method, and a Gaussian blur filter applied (1 μm radius). Time lapse corresponding to the mCherry-MLC channel were processed with the Particle Tracker function of the Mosaic plugin with the following settings: Cutoff 0.001, Per/Abs 0.6, Link Range 2, and Displacement 10. Only tracks shorter than 300 s were considered. A.csv file with tracks data, in particular track lenghts, was created and plotted by using the software GraphPad Prism.

**Western blotting.** For western blot analysis, cells were lysed directly on the plate by adding the opportune amount of modified Laemmli sample buffer composed of Tris·HCl 135 mM (pH 6.8), sodium dodecyl sulfate (SDS) 5%, urea 3.5 M, NP-40 2.3%, β-mercaptoethanol 4.5%, glycerol 4%, and traces of bromophenol blue. Total protein content was normalized by seeding cells at equal densities; this lysis buffer does not allow total protein quantification but prevents membrane-bound proteins from being degraded during trypsinization. The equal volume between samples was then loaded onto 12–8% SDS poly-acrylamide gels and transferred after electrophoretic separation onto a nitrocellulose membrane (Amersham GE-Healthcare). After the transfer, membranes were blocked in PBS supplemented with 0.1–0.3% Tween20 and 5% milk for 1 h at room temperature, then incubated overnight at 4° with primary antibodies at the following dilutions: mouse anti-βII-spectrin 1:2000 (BD-bioscience), rabbit anti-βII-spectrin 1:2000 (Abcam), and mouse anti-β-tubulin 1:5000 (Sigma-Aldrich). After three washing steps in PBS–Tween20 (0.1–0.3%), membranes were incubated for 1 h at room temperature with HRP-conjugated secondary antibodies (BioRad). Three washing steps of 5 min in PBS–Tween20 (0.1–0.3%) were performed between primary and secondary antibody incubation. Proteins were detected by ECL Western blotting reagents (Amersham GE-Healthcare), using the digital Chemidoc XRS + system run by the software Image Lab (Biorad).

**Model.** We proposed a model to investigate whether the spectrin periodic pattern seen between stress fibers in the cell emerges from a periodic hexagonal pattern, like that of the red blood cell. This hexagonal pattern better tils the 2D space and hence, facilitates the study of topological transitions. We hypothesized that this change in the cytoskeleton configuration is possible due to the interaction of the forces generated by its components. Therefore, to investigate the mechanical forces of the cell cytoskeleton, we modeled it as a net-work of springs and cables, as in refs. 34–39, where the edges and nodes correspond to filament bundles and cross-linkers, respectively. This mesoscopic description gives a good approximation for the spatio-temporal evolution of the cytoskeletal components under investigation. Unlike molecular dynamics and dissipative particle dynamics models[30,40], this approach allows the exploration of a larger cytoskeletal meshwork without computationally expensive simulations. Furthermore, we used a 2D network assuming that the out-of-plane deformations are negligible in the cells under consideration.

In the initial model, spectrin bundles were represented by the edges of a triangular mesh connected by short actin filaments (nodes). Spectrin bundles behave like springs, in the sense that they return to a resting length after stretching or shrinking. Hence, the $N_{e,S}$ spectrin bundles in our model generate a spring potential energy $U_{s,S}$ when they diverge from the resting length $d_{0,S}$, that is given by

$$U_{s,S} = \sum_{j=1}^{N_{e,S}} \frac{k_{s,S}\left(d_j - d_{0,S}\right)^2}{2}, \quad (2)$$

where $k_{s,S}$ is the spectrin spring constant and $d_j$ is the length of the edge $j$ expanding between the $l$ and $l'$ nodes.

To simulate the evolution of the spectrin mesh to a relaxed configuration where the potential energy generated by its edges mini-mizes, we assumed, as in refs. 35,36, that the position $\bar{r}_l \in \mathbb{R}^2$, $l \in \{1, \ldots, N_{n,A}\}$ of the nodes representing the actin short filaments connecting the spectrin bundles freely moves at a velocity $\bar{v}_l$, and that adhesion complexes generate a viscous resistance to cytoskeleton network deformations. In the model, the viscous resistance is given by $\xi \bar{v}_l$, where $\xi$ is a drag coefficient. This resistive force is balanced by the force generated by the spectrin potential energy at each node

$$\bar{F}(\bar{r}_l) = -\frac{\partial U_{s,S}}{\partial \bar{r}_l}. \quad (3)$$

Thus, $\bar{r}_l$ evolves according to

$$\frac{\partial \bar{r}_l}{\partial t} = \frac{\bar{F}_l}{\xi}. \quad (4)$$

The simulations were run in MATLAB_R2021a on a desktop computer. We used the delaunayTriangulation.m function to initialize the spectrin triangular mesh. Then, we solved Eq. 4 with the Euler method using small time-steps $\Delta t$ to ensure numerical stability. We evolved the system until there was no significant change in its configuration, i.e., until the system was "equilibrated". For Supplementary Fig. 8F the equilibration time is 60 s, for Fig. 6D–F 120 s, and for Fig. 6G–I 600 s. Since we were interested in the evolution between two different configurations of the spectrin mesh, instead of taking the mechanical parameters of the model from the literature related to one configuration or another, we fitted these parameters to qualitatively match the experimental observations. Other parameters, such as the length of the spectrin bundles, were taken from the literature. The units of the parameters were set to match the scales in the experiments. All the model parameters are in Supplementary Table 4.

In Fig. 6D, we introduced edges with a cable element in the model that produces a potential energy given by

$$U_c = \sum_{j=1}^{N_{e,c}} \frac{k_c d_j^2}{2}, \quad (5)$$

where $k_c$ is the cable constant. Note that the cable elements generate a force that shrinks the corresponding edges. Since they connect to the spectrin network on one end and points representing focal adhesions with zero velocity to the other end, these cable elements stretch the spectrin network. Note that the evolution of the network is now given by the balance of all the forces acting in the cell cytoskeleton, i.e.,

$$\frac{\partial \bar{r}_l}{\partial t} = \frac{\bar{F}_l}{\xi} = -\frac{\partial\left(U_{s,S} + U_c\right)}{\partial \bar{r}_l} \quad (6)$$

A recent study using molecular dynamics simulations[40] shows that an actin-spectrin model under a low strain rate is more prone to exhibit detachment of the actin-spectrin interface rather than fragmentation of the spectrin bundle. We included this observation by detaching the edges corresponding to spectrin bundles when an expanding force generated by their spring potential $U_{s,S}$ is greater than a force threshold $F_{th}$. We simulated the spectrin detachment from short actin filament by eliminating the corresponding edge from the network.

However, in the experimental data, the spectrin mesh was constrained by stress fibers. Therefore, we included stress fibers in the model by adding the corresponding edges to the top and the bottom of the spectrin mesh. As in ref. 36, these edges have a spring and a cable element with constants $k_{s,F}$ and $k_{c,F}$, respectively, creating a potential energy $U_F = U_{s,F} + U_{c,F}$. The contraction generated by the cable element resembles the contraction generated by the myosin rods along

the long actin filaments of the stress fibers. The stress fiber resting length $d_{0,F}$ corresponds to the calculated edge length in a relaxed state. Focal adhesions delimit stress fibers. The length of the edge connecting to focal adhesions was initially larger than the other edges in the stress fibers because we assumed that actin polymerization at the focal adhesions and actin exchange along the stress fiber also generate forces that affect the stress fibers resting state, as in ref. 41. Thus, this initial arrangement produces an instability that increases the length of all the stress fiber edges, thereby increasing their generated force and pulling the spectrin mesh to the focal adhesions. Moreover, during experiments, the stress fibers actively moved toward each other. To mimic this motion, the focal adhesion points moved vertically toward each other at a constant velocity $v_A$ during the first half of the simulation (300 s) and at zero velocity afterward.

We introduced myosin linkers connecting the spectrin mesh to the stress fibers in the model. The myosin linkers initially span from an extreme node of the stress fibers to the center of the closest triangle formed by spectrin bundles in the spectrin mesh. These edges have a cable element with constant $k_{c,L}$. Thus, they generate a shrinking force that affects the corresponding node in the stress fiber and the nodes of the spectrin triangle (the force is equally distributed among the three nodes of the spectrin triangle). We assumed that these myosin linkers do not shrink indefinitely, instead, they detach from the network if their lengths are less than $d_{min}$. When one of the sides of the spectrin triangle where a myosin linker connects detaches, then the myosin linker detaches from that spectrin triangle and attaches to a randomly selected free spectrin triangle within a radius $r$, $d_{min} < r < d_{max}$. If there are no free spectrin tringles, the myosin linker is removed from the network.

We also added myosin rods inside the spectrin mesh. These rods attach to spectrin triangles in both extremes and have a cable element with constant $k_{c,M}$. As the myosin linkers, if one of the sides of one spectrin triangle detaches, the corresponding extreme of the myosin rod randomly selects a free spectrin triangle within a radius $r$, $d_{min} < r < d_{max}$, to connect. If there are no free spectrin triangles within that radius, then the myosin rod is removed from the network. Furthermore, myosin rods are randomly created and removed from the spectrin mesh at a rate $\varphi_a$ and $\varphi_r$, respectively. This was implemented in the simulation by adding (removing) a myosin rod at each time-step if a randomly drawn number from a uniform distribution $u \sim U(0,1)$ is less than the probability $\Delta t \varphi_{a(r)}$. These rates and the initial number of myosin rods were informed by experimental observations.

The simulation of the full cytoskeleton network composed of edges representing spectrin bundles, stress fibers, myosin linkers or myosin rods, and nodes representing short actin filaments, focal adhesions, or stress fiber connectors, is described below:

At the start of the simulation:
- Initialize the network with the spectrin bundles, short actin filaments, stress fibers, focal adhesions, and myosin linkers.
- Add the myosin rods to the spectrin mesh to randomly selected locations, verifying that their length is within $d_{min}$ and $d_{max}$.
- At each time-step:

  - For each $j$ edge in the network spanning from $\bar{r}_l$ to $\bar{r}_{l'}$:
    - Calculate the force generated by the potential energy of their spring and/or cable elements $\bar{F}(\bar{r}) = -\frac{\partial U}{\partial r}$ at $\bar{r}_l$ and $\bar{r}_{l'}$.
    - Remove from the network an edge corresponding spectrin bundles if it generates a force larger than $F_{th}$.
  - Update the position of the nodes, by adding the force generated by their edges, $\bar{r} = \bar{r} + \frac{\bar{F}(\bar{r})}{\xi} \Delta t$.
    - For the focal adhesions: If time $< t_s$, move the nodes corresponding to the focal adhesions vertically at velocity $v_A$, hence $\bar{r} = \bar{r} + v_A \Delta t$. Otherwise, set the velocity to zero.
    - For the edges corresponding to myosin (rods or linkers) attached to a spectrin triangle that lost one of its sides: find a

new free spectrin triangle to attach within a radius $d_{min} < r < d_{max}$. If there are no free spectrin triangles within that radius, remove the spectrin edge from the network.
- If $u \sim U(0,1) < \Delta t \varphi_a$, add a myosin rod to a random location in the spectrin mesh.
- If $u \sim U(0,1) < \Delta t \varphi_r$, randomly select a myosin rod and remove it from the network.

In the model, the short F-actin position is dictated by the balance of forces generated by the connected spectrin bundles. Hence, there is feedback from spectrin to short F-actin by design: as long as spectrin bundles are connected to short F-actin, they will influence its position. Moreover, there must be feedback from the spectrin network to the stress fibers. As shown in Supplementary Fig. 10, the interaction between spectrin and stress fibers is important for network clustering. When the focal adhesions are fixed throughout the simulations, fewer spectrin bundles are detached, which hinders the transition to the four neighbor configuration (Supplementary Fig. 10I, M). Even changing the velocity at which the focal adhesions are pulled together can have large effects. For example, if the focal adhesions are faster, the spectrin network fails to compress at that rate, and it extends beyond the stress fibers, contrary to what is observed in experiments (Supplementary Fig. 10L). Therefore, there must be feedback between the spectrin network and the stress fibers to achieve an efficient spectrin network compression. Since spectrin network compression is achieved by the action of myosin rods and stress fiber dynamics are linked to the focal adhesion movement, we conclude that the morphological transitions are the result of this complex interaction.

## Reporting summary

Further information on research design is available in the Nature Portfolio Reporting Summary linked to this article.

## Data availability

The authors declare that all the original microscopy data that support the findings of this study are available from the corresponding authors upon request. All datasets used to generate each of the graphs presented in the work are included in the files accompanying this paper. Source data are provided with this paper.

## Code availability

The computational modeling codes can be found in the Github repository RangamaniLabUCSD/Spectrin-topological-transition [https://doi.org/10.5281/zenodo.10835918].

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

## Acknowledgements

We thank Paul Tillberg (HHMI Janelia) for the helpful discussion and advice on Expansion Microscopy, and Atsushi Fukuzawa (King's College London) for the molecular cloning of the FRET sensor. We are grateful to the IFOM imaging facility personnel, in particular D. Parazzoli, S. Magni, and E. Martini, for technical support. IFOM cell culture facility personnel for the help with the expansion of clonal population of *sptbn1* KO MEFs, and the maturation of hiPSC NGN2 cortical neurons. We thank all the members of Gauthier's, Scita's, and Maiuri's groups for their helpful discussion. This work was supported by the following funding: an Italian Association for Cancer Research (AIRC) Investigator Grant (IG) 27101 to N.C.G., by H2020-MSCA individual fellowship (796547) and Fondazione Cariplo Young Investigator Grant (2021-1507) to A.G., NIH Grant Number 1RF1DA055668-01 and by an Air Force Office of Scientific Research Grant FA9550-18-1-0051 to P.R. Additional funding by Fondazione Umberto Veronesi (FUV) doctoral fellowship to C.G. and "MilanoMarathon-oggicorroperAIRC" doctoral fellowship to M.C. (22461). C.G. and M.C. were PhD students at the European School of Molecular Medicine (SEMM).

## Author contributions

A.G. and N.C.G. designed the experimental study, and A.G. performed the experiments. M.B.Q. and P.R. conceived the computational model and M.B.Q. performed the in silico experiments. Z.L. performed STED microscopy, M.C. contributed to the microscopy analysis, and C. G. generated the genome-edited MEFs. A.G., M.B.Q., P.R., and N.C.G. wrote the manuscript. All authors read the manuscript and provided input.

## Competing interests

The authors declare no competing interests.
