## [Peer Review File · Nature Communications]

Mechanically induced topological transition of spectrin regulates its distribution in the mammalian cell cortexEditorial Note: This manuscript has been previously reviewed at another journal. This document only contains reviewer comments and rebuttal letters for versions considered at Nature Communications.

REVIEWER COMMENTS

Reviewer #1 (Remarks to the Author):

This is the third time I've seen this manuscript. It has gone through several significant efforts at improvement. I am satisfied that the authors have made serious efforts to listen to the concerns of the reviewers and qualify their statements about experimental approaches and applicability across different cell types.

The results presented in this paper may be applicable to other cells and provide an interesting view from which to consider the dynamic properties of spectrin-based structures.

I support publication now.

Reviewer #2 (Remarks to the Author):

Ghisleni et al. 2024 – The mechanically induced topological transition of spectrin regulates its distribution in the mammalian cortex

Spectrin-actin meshworks play structural and functional roles in erythrocytes and neurons, where they form triangular lattices and periodic structures, respectively. However, it is unclear how spectrin mesoscale cortical domains are organized in other cell types, as well as their mechanical properties and specialized functions. In this paper, Ghisleni et al. characterize the spatial organization and dynamics of spectrin clusters in fibroblasts. These spectrin clusters appear to be spatially segregated from actin stress fibers but still align according to the direction dictated by F-actin. Using expansion microscopy, the authors suggest that spectrin clusters assume two different ultrastructures: a diffusive mesh and a periodic constrained mesh. The authors suggest that these topologies are dynamic and dependent on myosin activity and contractility, with spectrin clusters providing a mechanoadaptive mechanism. Of particular interest is how periodic constrained spectrin clusters are not only sensitive to myosin activity but also actin stabilization. Experiments with FRET sensors reveal how spectrin tension is lower in periodic clusters. Finally, the authors propose a model to explain how spectrin can transition between the two topologies and introduce myosin as a main element driving the formation of spectrin periodic networks.

The authors have now addressed several of my points from the previous revision, especially concerning the lack of quantifications which have been substantially improved. The description of a periodic constrained mesh in fibroblasts is of interest to the field but the current data does not support the existence of a diffusive mesh, challenging the validity of the model and the conclusions that one can obtain from it.

Discussion of revision and remaining concerns:

1. Proper use of terminology.

- The authors have addressed my concerns and replaced ambiguous expressions such as “triangular erythroid-like configuration” or “erythroid-like lattice” with more adequate expressions that do not suggest erythroid-like configurations.

- I agree that the clusters fenced between stress fibers exhibit a periodic organization as this is made clear by the data in Figure 2D-F and Figure 4D. This observation is robust and has been a key point since the first submission. Moreover, the authors have added a sentence on the “Limitations to the Study” on how further work is required to directly address the transitions of spectrin between topological organizations. Concerning this, the authors have acknowledged in the manuscript how blebbistatin treatment promotes “cluster dissipation” (lines 250-251) instead of “favoring the diffused mesh” as they have mentioned in the point-by-point cover letter.

- The authors have replaced all instances of the term “condensate” and its derivatives with “clusters” and its derivatives, avoiding further confusion.

1. Model

- The model presented in Figures 6 and 7 aims to explain the transition between different spectrin topologies driven by force and myosin contractility which trigger spectrin unbinding. However, the model assumes the existence of a diffusive meshwork that transitions to a periodic, constrained meshwork; as explained in the “Main Issues” point, this network is not supported by the expansion microscopy images and subsequent quantification.

2. Lack of connection between in vivo and ExM datasets

- The authors have addressed this by providing further quantifications on signal autocorrelation for the ExM dataset as well as cluster area after jasplakinolide/blebbistatin treatments and during blebbistatin washouts. The new quantification plots strengthen the connection between jasplakinolide perturbation and transition to a periodic network, as well as the role of blebbistatin in promoting the dissipation of spectrin clusters.

3. Lack of quantification

- The authors have added autocorrelation plots for Jasplakinolide and Blebbistatin-treated cells (Figure S5C) as well as histograms for all the additional drug treatments (Figure S6B).

- Regarding the quantification of MLC pulse length, the authors have clarified that this has already been performed in Figure 7E. I would appreciate further detail on how many pulses have been quantified and how many cells have been analyzed for this plot.

- Upon my request, the authors have now provided additional analysis (Figure S9B-G) on MLC pulses in actin vs spectrin-rich regions of the cortex, highlighting a higher correlation between spectrin and high-intensity MLC pulses in spectrin-rich regions (Figure S9B-E). Moreover, the authors have also observed increased MLC/spectrin correlation during blebbistatin washout (Figure S9F-G). Both observations strengthen the connection between MLC pulses and spectrin clustering as suggested in Figure 7. To further expand on this hypothesis, the authors could also quantify the length of MLC pulses in spectrin vs actin-rich regions in the same datasets.

Remaining main issues

1. Although the authors no longer claim that spectrin exhibits an erythrocyte-like meshwork in fibroblasts, the manuscript still refers to a diffusive spectrin meshwork in fibroblasts. The authors refer to this meshwork as consisting of a polygonal spectrin topology with spectrin filaments connecting short actin nodes (Figure 2G). This meshwork is also represented in the theoretical model (Figures 6 and 7) to explain topological transitions driven by force and myosin contractility. However, this diffused meshwork still does not appear robust enough to be considered a different spectrin topology. Moreover, there is no visible observation of short actin filaments connecting spectrin filaments within any of the meshworks (diffused or constrained). Previous electron microscopy observations have identified a clear spectrin meshwork in erythrocytes (Liu et al, JCB 1987, Ursitti et al, Cell Motility and the Cytoskeleton 1991) which is more regular and widespread compared to the diffusive mesh presented in the manuscript. Therefore, to validate this point, the authors could perform EM imaging of spectrin in unroofed fibroblasts. Alternatively, the authors could quantify the distribution of actin within the diffused spectrin meshwork and perform dual-color expansion microscopy imaging of N-Term/C-Term β II-Spectrin or the capping protein adducin. Similar observations were done in erythrocytes with 1-color/2-color STORM (Pan et al., Cell Reports 2018) and contributed to establishing a more robust observation of the spectrin-actin meshwork. Concerning this, the authors have quantified the spectrin lattice in erythrocytes with results comparable to previous super-resolution studies (Pan et al., Cell Reports 2018); however, the length of the spectrin filaments is 2-fold lower than the reported for the diffusive mesh (92 vs 215). This could also suggest that the diffusive mesh does not constitute a specific spectrin topology or that spectrin adopts multiple tensional states when not constrained into periodic clusters. Without a more systematic characterization of the diffusive mesh, it is unclear whether spectrin transitions between different topologies.

2. If the diffusive mesh does not appear robust after the experiments suggested on the previous point, or these are considered out of the scope of the current manuscript, the authors could modify the message and explain how spectrin transitions from a heterogeneous, unordered state to a periodic constrained mesh. This is an important finding for the field and in my opinion enough for publication but would require a change of the theoretical model and a possible discussion on why spectrin assumes different topologies in different cell types.

Response to REVIEWER COMMENTS

Reviewer #1 (Remarks to the Author):

This is the third time I've seen this manuscript. It has gone through several significant efforts at improvement. I am satisfied that the authors have made serious efforts to listen to the concerns of the reviewers and qualify their statements about experimental approaches and applicability across different cell types.

The results presented in this paper may be applicable to other cells and provide an interesting view from which to consider the dynamic properties of spectrin-based structures.

I support publication now.

We thank the reviewer for improving the quality of our manuscript and support the publication.

Reviewer #2 (Remarks to the Author):

Ghisleni et al. 2024 – The mechanically induced topological transition of spectrin regulates its distribution in the mammalian cortex
Spectrin-actin meshworks play structural and functional roles in erythrocytes and neurons, where they form triangular lattices and periodic structures, respectively. However, it is unclear how spectrin mesoscale cortical domains are organized in other cell types, as well as their mechanical properties and specialized functions. In this paper, Ghisleni et al. characterize the spatial organization and dynamics of spectrin clusters in fibroblasts. These spectrin clusters appear to be spatially segregated from actin stress fibers but still align according to the direction dictated by F-actin. Using expansion microscopy, the authors suggest that spectrin clusters assume two different ultrastructures: a diffusive mesh and a periodic constrained mesh. The authors suggest that these topologies are dynamic and dependent on myosin activity and contractility, with spectrin clusters providing a mechanoadaptive mechanism. Of particular interest is how periodic constrained spectrin clusters are not only sensitive to myosin activity but also actin stabilization. Experiments with FRET sensors reveal how spectrin tension is lower in periodic clusters. Finally, the authors propose a model to explain how spectrin can transition between the two topologies and introduce myosin as a main element driving the formation of spectrin periodic networks.
The authors have now addressed several of my points from the previous revision, especially concerning the lack of quantifications which have been substantially improved. The description of a periodic constrained mesh in fibroblasts is of interest to the field but the current data does not support the existence of a diffusive mesh, challenging the validity of the model and the conclusions that one can obtain from it.

Discussion of revision and remaining concerns:

1. Proper use of terminology.

- The authors have addressed my concerns and replaced ambiguous expressions such as "triangular erythroid-like configuration" or "erythroid-like lattice" with more adequate expressions that do not suggest erythroid-like configurations.
- I agree that the clusters fenced between stress fibers exhibit a periodic organization as this is made clear by the data in Figure 2D-F and Figure 4D. This observation is robust and has been a key point since the first submission. Moreover, the authors have added a sentence on the "Limitations to the Study" on how further work is required to directly address the transitions of spectrin between topological organizations. Concerning this, the authors have acknowledged in the manuscript how blebbistatin treatment promotes "cluster dissipation" (lines 250-251) instead of "favoring the diffused mesh" as they have mentioned in the point-by-point cover letter.
- The authors have replaced all instances of the term "condensate" and its derivatives with "clusters" and its derivatives, avoiding further confusion.

We would like to thank the reviewer for improving the quality and clarity of our manuscript. We acknowledge that some statements included in early versions of this manuscript may have led to controversial over-interpretation of the results. On this premise, we are submitting a revised version of the manuscript that emphasizes the key and novel observation of the periodic topology of spectrin clusters, rather than the transition between different configurations.

1. Model

- The model presented in Figures 6 and 7 aims to explain the transition between different spectrin topologies driven by force and myosin contractility which trigger spectrin unbinding. However, the model assumes the existence of a diffusive meshwork that transitions to a periodic, constrained meshwork; as explained in the "Main Issues" point, this network is not supported by the expansion microscopy images and subsequent quantification.

We agree with the reviewer that our experiments do not give robust evidence of erythrocyte diffusive mesh. However, to observe topological transitions to a cluster state in our model, we need to start from a diffusive configuration. Selecting any random initial configuration could lead to inconclusive results due to the stochasticity of our model. Hence, we selected an idealized triangular configuration, which is the best way to fill the 2D space. Moreover, it is isotropic, which avoids initial bias to a clustered state. In our previous manuscript, we linked this triangular configuration to a red blood cell configuration. Now, addressing the reviewer's concern, we have changed the text (changes are highlighted in red in the revised manuscript) to justify the initial configuration to geometrical arguments, avoiding any reference to the configuration of red blood cells. This issue is now also addressed in the Limitation to the study.

2. Lack of connection between in vivo and ExM datasets

- The authors have addressed this by providing further quantifications on signal autocorrelation for the ExM dataset as well as cluster area after jasplakinolide/blebbistatin treatments and during blebbistatin washouts. The new quantification plots strengthen the connection between jasplakinolide perturbation and transition to a periodic network, as well as the role of blebbistatin in promoting the dissipation of spectrin clusters.

We are pleased the reviewer now supports our data presentation.

3. Lack of quantification

- The authors have added autocorrelation plots for Jasplakinolide and Blebbistatin-treated cells (Figure S5C) as well as histograms for all the additional drug treatments (Figure S6B).
- Regarding the quantification of MLC pulse length, the authors have clarified that this has already been performed in Figure 7E. I would appreciate further detail on how many pulses have been quantified and how many cells have been analyzed for this plot.

The number of pulses originally presented were ~2000 for the cell displayed. We have further expanded the number of events and cells analyzed as specified in the figure legends.

- Upon my request, the authors have now provided additional analysis (Figure S9B-G) on MLC pulses in actin vs spectrin-rich regions of the cortex, highlighting a higher correlation between spectrin and high-intensity MLC pulses in spectrin-rich regions (Figure S9B-E). Moreover, the authors have also observed increased MLC/spectrin correlation during blebbistatin washout (Figure S9F-G). Both observations strengthen the connection between MLC pulses and spectrin clustering as suggested in Figure 7. To further expand on this hypothesis, the authors could also quantify the length of MLC pulses in spectrin vs actin-rich regions in the same datasets.

We believe figure 7B-C exactly point to the longer lifetime of MLC tracks on stress fiber vs MLC pulses in other cell regions. In the limitation to the study, we address the need of further improving spatial and temporal resolutions of live dataset, including myosin dynamics.

Remaining main issues

1. Although the authors no longer claim that spectrin exhibits an erythrocyte-like meshwork in fibroblasts, the manuscript still refers to a diffusive spectrin meshwork in fibroblasts. The authors refer to this meshwork as consisting of a polygonal spectrin topology with spectrin filaments connecting short actin nodes (Figure 2G). This meshwork is also represented in the theoretical model (Figures 6 and 7) to explain topological transitions driven by force and myosin contractility. However, this diffused meshwork still does not appear robust enough to be considered a different spectrin topology. Moreover, there is no visible observation of short actin filaments connecting spectrin filaments within any of the meshworks (diffused or constrained). Previous electron

microscopy observations have identified a clear spectrin meshwork in erythrocytes (Liu et al, JCB 1987, Ursitti et al, Cell Motility and the Cytoskeleton 1991) which is more regular and widespread compared to the diffusive mesh presented in the manuscript. Therefore, to validate this point, the authors could perform EM imaging of spectrin in unroofed fibroblasts. Alternatively, the authors could quantify the distribution of actin within the diffused spectrin meshwork and perform dual-color expansion microscopy imaging of N-Term/C-Term β II-Spectrin or the capping protein adducin. Similar observations were done in erythrocytes with 1-color/2-color STORM (Pan et al., Cell Reports 2018) and contributed to establishing a more robust observation of the spectrin-actin meshwork. Concerning this, the authors have quantified the spectrin lattice in erythrocytes with results comparable to previous super-resolution studies (Pan et al., Cell Reports 2018); however, the length of the spectrin filaments is 2-fold lower than the reported for the diffusive mesh (92 vs 215). This could also suggest that the diffusive mesh does not constitute a specific spectrin topology or that spectrin adopts multiple tensional states when not constrained into periodic clusters. Without a more systematic characterization of the diffusive mesh, it is unclear whether spectrin transitions between different topologies.

2. If the diffusive mesh does not appear robust after the experiments suggested on the previous point, or these are considered out of the scope of the current manuscript, the authors could modify the message and explain how spectrin transitions from a heterogeneous, unordered state to a periodic constrained mesh. This is an important finding for the field and in my opinion enough for publication but would require a change of the theoretical model and a possible discussion on why spectrin assumes different topologies in different cell types.

We fully understand the concern expressed by the reviewer and the lack of a clear distinct topological organization of the Diffused mesh. Given the complexity of the technical approaches suggested, and the uncertainties of platinum replica EM in visualizing spectrin *in situ* (see Vassilopoulos *et al.*, 2019, Nat Comm), we have decided to reframe the manuscript as suggested in the point 2 (the changes are highlighted in red in the revised manuscript and figures 2G-5D-7I have been modify to remove any potential confusing reference to a triangular mesh in fibroblasts). We are now focusing our data on the spectrin transition from a non-polarized heterogeneously diffused mesh toward the periodic pattern, as observed in the clusters by ExM. We hope the reviewer appreciates this convergence point and supports the publication of what we all agree being "an important finding for the field".

REVIEWERS' COMMENTS

Reviewer #2 (Remarks to the Author):

I now agree with this reviewed version of the manuscript. In my opinion it is now much clearer and the conclusions are supported by the data. I fully support publication.